# Unsupervised Learning of Equivariant Structure from Sequences

**Takeru Miyato**[*1,2] **Masanori Koyama**[*1] **Kenji Fukumizu**[3,1]   [*]equal contribution
[1]Preferred Networks, Inc. [2]University of Tübingen [3]The Institute of Statistical Mathematics

## Abstract

In this study, we present *meta-sequential prediction* (MSP), an unsupervised framework to learn the symmetry from the time sequence of length at least three. Our method leverages the stationary property (e.g. constant velocity, constant acceleration) of the time sequence to learn the underlying equivariant structure of the dataset by simply training the encoder-decoder model to be able to predict the future observations. We will demonstrate that, with our framework, the hidden disentangled structure of the dataset naturally emerges as a by-product by applying *simultaneous block-diagonalization* to the transition operators in the latent space, the procedure which is commonly used in representation theory to decompose the feature-space based on the type of response to group actions. We will showcase our method from both empirical and theoretical perspectives. Our result suggests that finding a simple structured relation and learning a model with extrapolation capability are two sides of the same coin. The code is available at https://github.com/takerum/meta_sequential_prediction.
.

## 1   Introduction

The recent evolution and successes of neural networks in machine learning fields have shown the importance of symmetry-aware neural network models [18, 45, 38, 63]. In particular, symmetries in the form of geometric/algebraic constraints have been proven useful in various applications involving high-dimensional, highly-structured observations. For example, recent literature of robotics and reinforcement learning has succeeded in exploiting the knowledge of geometrical symmetries to improve the sample efficiency [62, 65] or to train a model that generalizes to unseen observations [58].

However, building an inductive bias that matches the given dataset of interest is challenging, and recent studies have been exploring the ways to learn symmetries itself from observational sequences. Many of these approaches consider settings with relatively restrictive assumptions or weak supervision. For example, [56] allows the trainer to use the knowledge of the identities of the actions used in making the transition. Meanwhile, [4, 35, 34] essentially assume that the transition velocity of all sequences in the datasets are the same.

These studies indicate that there is still much room left for the question of "what is required for dataset/model to enable the unsupervised learning of the equivariance relation corresponding to the underlying symmetry". This paper advances this investigation by showing that if the sequential dataset consists of time series with a certain stationary property (constant velocity, constant acceleration), we can learn the underlying symmetries by simply training the model to be able to predict the future with linear transition in the latent space. Our theory in Section 3 shows that this strategy can learn a model that is almost equivariant. Moreover, we will experimentally show that, by training an encoder-decoder model in a framework of meta-learning which we call **meta-sequential prediction** (MSP), we can actually learn an equivariant model. In particular, we show that we can

36th Conference on Neural Information Processing Systems (NeurIPS 2022).

learn a hidden equivariance structure in the dataset by splitting (1) the internal training step to compute the prediction loss of linear transition in the latent space from (2) the external training step to update the encoder and decoder. We will also empirically show that, in alignment with group representation theory [42], the learned linear latent transitions in our framework can be simultaneously block-diagonalized, and that each block corresponds to a disentangled factor of variation.

## 2 Related works

Recently, numerous studies have explored the ways to learn symmetry in a data-driven manner. There is rich literature in unsupervised/weakly supervised approaches that use sequential datasets to exploit the structure that is shared across time, and they all differ by the types of inductive bias. For example, the object-centric approach introduces inductive bias in the form of architecture, and equips the model with pre-defined slots to be allocated to objects [39, 33, 40]. Meanwhile, [22, 1] assumes that the symmetry to be found takes the form of the energy conservation law, and learns each variable in the law as a function of observations. While this approach assumes that some *energy* is preserved in each observation, we assume that the transition parameters like velocity and acceleration are preserved within each observation. Other more indirect forms of inductive bias include those relevant to distributional sparseness and symmetry defined through algebraic constraints. [31] for instance assumed that every stationary component of a given time series is generated by a finite and independent latent time series. [41] proposed to sparsely model the transition with a distribution of large kurtosis. Our work belongs to a family of unsupervised learning that seeks to find the underlying symmetry of the dataset based on an algebraic inductive bias that the transitions can be represented linearly in some latent space. In this sense, our inductive bias also has a connection to Koopman operator [43, 25, 3]. We are different from these studies in that we are aiming to learn a common encoding function (i.e. lifting function) under which the *set* of sequences following different dynamics can be described linearly. Also [70] applied Koopman operator on pedestrian walking sequences, and [23] used Koopman operator to separate the foreground from the background. However, [70] does not set out their model to solve the extrapolation, and neither [70] nor [23] discusses the natural algebraic decomposition of the latent space that results solely from the objective to predict the unseen future.

**Unsupervised learning with algebraic/geometrical constraints**   Many studies impose algebraic constraints that reflect some form of geometrical assumptions. [16] uses a known coordinate map parametrization of a Lie group family to construct a posterior distribution on the manifold. [54] assumes that the observations are dense enough on the data-manifold to describe its tangent space, and exploits a property of random walks on the product manifold to decompose the data space. In the analysis of sequential datasets, [13, 59, 10, 56, 12] make some Lie group type assumptions about the transition. [56] also assumes that the identity of the actions in the sequences is known. As for the approaches with less explicitly geometrical touch, [35] uses capsule structure in their probabilistic framework to model a finitely cyclic structure while retaining the computability of posterior distribution. By design, [35] assumes that all sequences in the dataset transition with the same cyclic velocity. [69] enforces the underlying transition action to be commutative. Some of these studies learn the representation so that the linear transition in the latent space can be explicitly computed [56, 12, 4]. In particular, [4] presents a theory that suggests that a representation without this feature would have topological defects, such as discontinuity. Our approach shares a similar philosophy with these works except that, instead of imposing a strong assumption about the underlying symmetry, we only make a relatively weak stationarity assumption about the dataset; although we assume each sequence to be transitioning with constant velocity/acceleration, we allow the velocity/acceleration to vary across different sequences.

**Disentangled Representation Learning**   Disentangled structure [29, 30] is a form of symmetry that has also been actively studied. It is known that, under the i.i.d assumption of examples, *unsupervised* learning of disentanglement representation is not achievable without some inductive biases encoded in models and datasets [49]. In response to this work, subsequent works have explored different frameworks such as weakly-/semi-supervised settings [51, 50] and learning on sequential examples [41] to learn disentangled representations. For example, PhyDNet [24] disentangles the known physical dynamics from the unknown factors by preparing an explicit module called PhyCell. ICA [32] and recent works [71, 64] also discuss the identifiability property of learned representa-

tions. Classical methods like [28, 6, 36] take an approach of incorporating the inductive bias in the form of a probabilistic model.

We are different from many previous methods in that we do not equip our model with an explicit disentanglement framework. Our method achieves disentanglement as a by-product of training a model that can predict the future linearly in the latent space. The set of latent linear transformations estimated by our method for different time sequences can be simultaneously block-diagonalized, and the latent space of each block corresponds to a disentangled feature. Our data assumption about constant velocity/acceleration might be similar in taste to the setting used by [32], in which the observed time series can be split into the finite number of stationary components.

## 3 Learning of equivariant structure from stationary time sequences

Our goal is to learn the underlying symmetry structure of a dataset in an unsupervised way that helps us predict the future. What do us humans require for the dataset when we are tasked to, for example, predict where a thrown ball would be in the next second? We hypothesize that we solve such a prediction task by analyzing a short, past time-frame with a certain stationary property (e.g., constant velocity/acceleration). Indeed, people with good dynamic visual acuity can chase a fast-moving object, because they can identify such a short stationary time-frame and use it to predict the near future *linearly* in their latent space. Based on this intuition, we propose to provide the trainer with a dataset consisting of constant velocity/acceleration sequences. We formalize this idea below.

**Dataset structure**  Our dataset $\mathcal{S}$ consists of sequences in some ambient space $\mathcal{X}$, so that each member $\mathbf{s} \in \mathcal{S}$ takes the form $\mathbf{s} = [s_t \in \mathcal{X}; t = 1, ..., T] \in \mathcal{S}$. Because we want $\mathbf{s}$ to be describing a sequence that transitions with constant velocity, we assume that all $s_t$ in a given instance of $\mathbf{s} \in \mathcal{S}$ are related by a fixed transition operation $g \in \mathcal{G}$ so that $s_{t+1} = g \circ s_t$ for all $t$, where $\mathcal{G}$ is the set of transition operators on $\mathcal{X}$ and each $g \in \mathcal{G}$ acts on $x \in \mathcal{X}$ by sending $x$ to $g \circ x$. We assume that $\mathcal{X}$ is closed under $\mathcal{G}$; that is, $g \circ x \in \mathcal{X}$ for all $x \in \mathcal{X}, g \in \mathcal{G}$. We allow $\mathcal{G}$ to be continuous as well, so that $g$ might not have a finite order (For instance, if $g$ is a rotation with speed $2\pi r$ with irrational $r$, any finite repetition of $g$ would not agree with identity mapping). This way, our setting is different from those used in [35] that explores a cyclic structure using the capsules of same size. We emphasize that the transition action $g$ is generally assumed to differ across the different members of $\mathcal{S}$. For example, if $\mathcal{G}$ is a set of rotations and $\mathcal{X}$ a set of images, then the rotational speed, direction and the initial image may all be different for any two distinct sequences, $\mathbf{s}$ and $\mathbf{s}'$. Because each instance of $\mathcal{S}$ is characterized by $s_1$ and $g$, we may write $\mathbf{s}(s_1, g)$ to denote a sequence that begins with initial frame $s_1$ and transitions with $g$. Summarizing, $\mathcal{S}$ is a subset of $\{[g^t \circ s_1; t = 0, ..., T-1]; s_1 \in \mathcal{X}, g \in \mathcal{G}\}$.

**Prediction framework through equivariance**  Our strategy is to exploit the stationary property of each $\mathbf{s} \in \mathcal{S}$ to seek an invertible continuous function $\Phi : \mathcal{X} \to \mathbb{R}^{a \times m}$ such that there exists some $M : \mathcal{G} \to \mathbb{R}^{a \times a}$ satisfying

$$M_g \Phi(x) = \Phi(g \circ x) \text{ for all } x \in \mathcal{X} \text{ and } g \in \mathcal{G}. \tag{1}$$

This relation is known as equivariance [11], and this type of tensorial latent space has also been used in [44] as well for the unsupervised learning of the structure underlying the dataset. In other words, we seek a model in which $\mathcal{X}$ and $\mathbb{R}^{a \times m}$ are invertibly related by an equivariance relation with respect to $\mathcal{G}$, where $g \in \mathcal{G}$ acts on $\Phi(x) \in \mathbb{R}^{a \times m}$ via the map $\Phi(x) \mapsto M_g \Phi(x)$ with $M_g \in \mathbb{R}^{a \times a}$. We assume $m > 1$ in our study[1]. In this framework, we predict the sequence $\mathbf{s}(s_1, g)$ with the relation $\Phi^{-1}(M_g \Phi(s_{t-1})) = \tilde{s}_t$. When $\mathcal{G}$ is a group, (1) would imply $M_{gh}\Phi(x) = \Phi(gh \circ x) = M_g \Phi(h \circ x) = M_g M_h \Phi(x)$ for all $x$, and the map $g \mapsto M_g$ is called a *representation* of $\mathcal{G}$ [9, 67, 10].

Because we are aiming to establish the framework in which the representation of each action $g$ can be explicitly computed, our philosophy has much in common with the proposition of [4]. This approach is in contrast to [35], which encodes a predefined cyclic structure in the model.

---

[1] We note that, when $m > 1$, the action of $M_g$ on $\mathbb{R}^{a \times m}$ can be realized by applying the same $M_g$ to $m$-copies of $\mathbb{R}^a$. In other words, our prediction framework assumes that there are $m$ number of subspaces that react to $g$ in the same way. This $m > 1$ assumption is also considered in [4]. The case in which there are no copies of subspaces that act in the same way is called multiplicity-free in the literature of representation theory [9, 19], and is known to be a special case that happens only under a restrictive condition on the dimension of the observations space [46]. A similar idea has been used in model architecture as well [15].

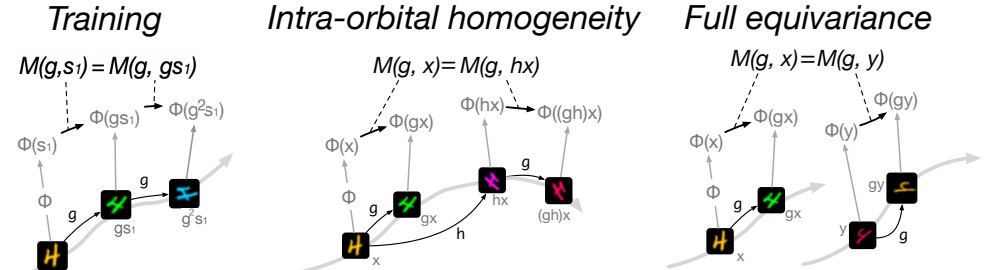

Figure 1: Visualization of intra-orbital homogeneity vs full equivariance. During the training, the model was trained to satisfy $M(g, s_1) = M(g, g \circ s_1)$ for all $g$ and $s_1$. When intra-orbital homogeneity holds, $M(g, x) = M(g, h \circ x)$ for all $h, g \in G$ and $x$. When the full equivariance holds, $M(g, x)$ is invariant across different orbits.

### 3.1 Learning equivariance relation from stationary sequential dataset

However, training the model satisfying (1) with just the *constant velocity assumption* is not a trivial task, because this model assumption only assures that, for each sequence $\mathbf{s}(s_1, g)$, there is a sequence-specific operator $M(g, s_1)$ that is guaranteed only to be able to predict the sequence that transitions with $g$ and begins from $s_1$ in the way of $M(g, s_1)\Phi(s_t) = \Phi(s_{t+1}) = \Phi(g \circ s_t)$ (the left most panel in Figure 1). In order to satisfy the *full* equivariance ((1) or the right most panel Figure 1), $M(g, s_1)$ shall not depend on $s_1$ (i.e. homogeneous with respect to $s_1$). At the same time, because the constant velocity assumption applies to each sequence over all time intervals, it at least assures that the latent transition $M$ is well defined within each sequence; that is, $M(g, x) = M(g, g \circ s_1) = M(g, g^2 \circ s_1) \cdots$ and so on. It turns out that, with some regularity assumptions on the model and the choice of $\mathcal{G}$, we can extend this observation to say that $M$ satisfies **intra-orbital homogeneity** (the middle panel in Figure 1) ; that is, $M(g, x)$ is constant on the orbit $\mathcal{G} \circ x = \{g \circ x; g \in \mathcal{G}\}$ for each $x$.

**Proposition 3.1.** *Suppose that $\Phi(s_t) = M(g, s_1)\Phi(s_{t-1})$ for all $\mathbf{s}$ and $t$. If $m > a$ and if $\mathcal{G}$ is a compact commutative Lie group, then $M$ satisfies intra-orbital homogeneity.*

Also, if $M$ satisfies intra-orbital homogeneity, $M(g, x)$ and $M(g, x')$ for any pair $(x, x')$ in different orbits $\mathcal{G} \circ x \neq \mathcal{G} \circ x'$ can be shown to be at least similar.

**Proposition 3.2.** *Suppose that $M(g, x)$ satisfies intra-orbital homogeneity, and suppose that $\mathcal{G}$ is a compact connected group. If $M(g, x)$ is continuous with respect $x$, then for all $(x, x')$, there exists some $P$ such that $PM(g, x)P^{-1} = M(g, x')$.*

Thus, much of the equivariance property can be satisfied automatically by training the representation on the set of stationary sequences. Interestingly, as we experimentally demonstrate later, our training method in the next section successfully learns a fully equivariant $\Phi$ without explicitly enforcing the change of basis $P$ to be $I$.

### 3.2 Learning $\Phi$ via solving a *meta-sequential prediction* task

We propose a meta-learning way to learn a homeomorphic function $\Phi : \mathcal{X} \to \mathbb{R}^{a \times m}$ with equivariance property by seeking an injective $\Phi$ such that $M(g, s_1)\Phi(s_t) = \Phi(s_{t+1})$ for all $g \in \mathcal{G}$, $s_1 \in \mathcal{X}$. We learn such $\Phi$ by casting this problem as a meta-learning problem in which $M(g, s_1)$ is to be internally estimated for each $\mathbf{s}$. In other words, we seek a pair of an encoder $\Phi$ and a decoder $\Psi$ such that $\mathcal{L}(\Phi, \Psi|\mathbf{s}) = \min_M \sum_{s_t, s_{t+1} \in \mathbf{s}} \|\Psi(M\Phi(s_t)) - s_{t+1}\|_2^2$ is optimized for each $\mathbf{s}$.

We conduct this optimization by splitting each $\mathbf{s} = \{s_1, ..., s_T\}$ into conditional time sequence $\mathbf{s}_c = \{s_1, ..., s_{T_c}\}$ and validation time sequence $\mathbf{s}_p = \{s_{T_c+1}, ..., s_T\}$, while using the former for the internal optimization of $M$ to force the linear algebraic relation in the latent space and using the latter for the prediction loss. More precisely, we solve the following optimization problem about $\Phi$

and $\Psi$:

$$\mathcal{L}^p(\Phi, \Psi) := \sum_{\mathbf{s}}\sum_{t=T_c+1}^{T} \left\| \Psi(M^*(\mathbf{s}_c|\Phi)^{t-T_c}\Phi(s_{T_c})) - s_t \right\|_2^2$$

$$\text{where } M^*(\mathbf{s}_c|\Phi) = \arg\min_M \sum_{t=1}^{T_c-1} \|M\Phi(s_t) - \Phi(s_{t+1})\|_F^2 . \tag{2}$$

Since $M^*$ is obtained from the latent sequence in the internal optimization, we call this learning framework the **meta-sequential prediction** (MSP). It might appear as if we can also set $\mathbf{s} = \mathbf{s}_c$ and optimize the following reconstruction version of Eq.(2):

$$\mathcal{L}^r(\Phi, \Psi) := \sum_{\mathbf{s}}\sum_{t=2}^{T_c} \|\Psi(M^*(\mathbf{s}|\Phi)^{t-1}\Phi(s_1)) - s_t\|_2^2. \tag{3}$$

However, as we will see in the experiment section, the use of the validation sequence $\mathbf{s}_p$ makes a substantial difference in the learned representation. This is most likely because the minimization of the validation error of $M^*$ on $\mathbf{s}_p$ would encourage $\Phi$ to exclude the $\mathbf{s}_c$-specific information from the transition $M^*$. We will illustrate this effect in the experiment section. We shall note that we can also parameterize $M$ as $M := \exp(A)$, where $A \in \mathbb{R}^{a \times a}$ is a Lie algebra element to be internally optimized. This type of approach was used in [14] for building Lie group convolutional network and in [59, 12] for predicting a sequence that is not necessarily stationary. In order to train their model, [59] used additional parameters to diagonalize each algebra element as well as hyperparameters to stabilize the training. We also experimented with a Lie-algebra style representation of $M^*$ and used SGD to internally optimize the exponent parameters, but we needed to carefully tune the hyperparameter for the norm regularization term to stabilize the training and never really succeeded to train the model without collapsing. Our internal optimization procedure is free of such a parameter tuning.

**Internal Optimization of $M^*$** Because the internal optimization in Eq.(2) is a linear problem, it can be solved analytically as

$$M^*(\mathbf{s}_c|\Phi) = H_{+1}H_{+0}^\dagger, \tag{4}$$

where $H_{+0} = [\Phi(s_1); ...; \Phi(s_{T_c-1})] \in \mathbb{R}^{a \times (T_c-1)m}$ and $H_{+1} = [\Phi(s_2); ...; \Phi(s_{T_c})] \in \mathbb{R}^{a \times (T_c-1)m}$ are the horizontal concatenations of the encoded frames and $H_{+0}^\dagger$ is the Moore-Penrose pseudo inverse of $H_{+0}$. Because $M^*(\mathbf{s}_c|\Phi)$ is a closed form with respect to $\Phi$, the loss (2) can be directly optimized by differentiating it with respect to the parameters of both $\Phi$ and $\Psi$. Thus, the training is done in an end-to-end manner. Figure 2 summarizes the overall procedure to make prediction on a given sequence when $T_c = 2, T_p = t$. We note that,

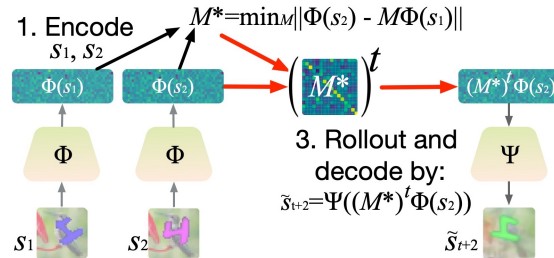

Figure 2: The overview of *meta-sequential prediction* (MSP) when $T_c = 2$ and $T_p = t$. After the encoder encodes the observations into tensor representations $\Phi(s_1), \Phi(s_2)$, the method solves the least square problem: $M^* = \arg\min_M \|M\Phi(s_1) - \Phi(s_2)\|_F^2$. The model then predicts the future observations by $\tilde{s}_{t+2} = \Psi((M^*)^t\Phi(s_2))$. These processes (including the linear problem) are all differentiable.

although we have assumed the dataset to consist of constant-velocity sequences, we can readily extend our method to the dataset consisting of the time series with higher-order stationarity, such as constant acceleration. See Section 4.4 for the detailed explanation of the model extension and the experimental results.

### 3.3 Irreducible decomposition of $M^*$s

Representation theory guarantees that, if $\mathcal{G}$ is a compact connected group, any representation $D : \mathcal{G} \to \mathbb{R}^{a \times a}$ can be simultaneously block-diagonalized; that is, there is a common change of basis $U$ such that $V := UD_gU^T = \bigoplus_j V_g^{(j)}$, where $V_g^{(j)}$ is called irreducible representation that cannot be block-diagonalized any further [42, 10, 67]. The equivariance of our $\Phi$ which we show in the experimental section suggests that $M^*(\mathbf{s}|\Phi)$ may be simultaneously block-diagonalizable as well. This block-diagonalization sometimes reveals disentanglement structure because any irreducible representation of $\mathcal{G}_1 \times \mathcal{G}_2$ is of form $V^{(1)} \otimes V^{(2)}$, where $V^{(k)}$ is an irreducible representation of

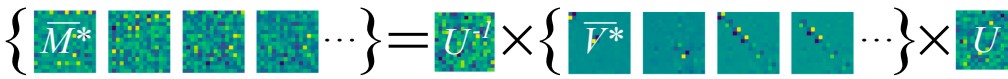

Figure 3: Simutaneous block diagonalization (SBD) applied to the set of $M^*$s obtained from 3DShapes sequences. SBD finds the common change of basis under which all matrices $V^* = UM^*U^{-1}$ simultaneously take the form of block diagonal matrices with the same block positions. For clarity, we provide in this figure the visualizations of $\overline{M^*} := M^* - I$ and $\overline{V^*} := V^* - I$ instead of $M^*$ and $V^*$.

$\mathcal{G}_k$. In particular, if $M^*$'s irreducible representations have the form $V^{(1)} \otimes 1$ or $1 \otimes V^{(2)}$, then each block would either corresponds to the action of $\mathcal{G}_1$ or of $\mathcal{G}_2$.[2]

To find $U$ that simultaneously block-diagonalizes all $M^*(\mathbf{s}|\Phi)$, we optimized $U$ based on the following objective function that measures the block-ness of $V^*(\mathbf{s}) := UM^*(\mathbf{s}|\Phi)U^{-1}$ based on the normalized graph Laplacian operator $\Delta$:

$$\mathcal{L}_{\mathrm{bd}}(V^*(\mathbf{s})) := \|\Delta(A(V^*(\mathbf{s})))\|_{\mathrm{trace}} = \sum_{d=1}^{a}\sigma_d(A(V^*(\mathbf{s}))) \tag{5}$$

where $A(V^*(\mathbf{s})) = abs(V^*(\mathbf{s}))abs(V^*(\mathbf{s}))^{\mathrm{T}}$ with $abs(V^*(\mathbf{s}))$ representing the matrix such that $abs(V^*(\mathbf{s}))_{ij} = |V^*_{ij}|$. Our objective function is based on the fact that, if we are given an adjacency matrix $A$ of a graph, then the number of connected components in the graph can be identified by looking at the rank of the graph Laplacian. For the derivation, please see Appendix E. Through this decomposition, we are able to uncover the hidden block structure of $M^*$s. See Figure 3 for the actual block-decompositions of $M^*$s through our simultaneous block diagonalization. We show in Section 4.3 that each block component of $V^*$ with optimized $U$ corresponds to the disentangled factor of variations in dataset.

## 4 Experiments

We conducted several experiments to investigate the efficacy of our framework. In this section, we briefly explain the experimental settings. For more details, please see Appendix D. We tested our framework on *Sequential* MNIST, 3DShapes [5], and SmallNORB [48]. Sequential MNIST is created from MNIST dataset [47]. For all experiments, we used a ResNet [26]-based encoder-decoder architecture and we set $a = 16$ and $m = 256$ so that the latent space lives in $\mathbb{R}^{16 \times 256}$.

For Sequential MNIST, we chose our $\mathcal{G}$ to be the set of all combinations of three types of transformations: shape rotation, hue rotation, and translation, and randomly sampled a single instance of $g \in \mathcal{G}$ for each sequence(See Appendix D for the examples of sequences). To create each sequence, we first resized the MNIST image to 24×24, applied repetitions of a randomly sampled, fixed member of $g \in \mathcal{G}$ and embedded the results to $32 \times 32$ images. For shape and hue rotations, we randomly sampled the velocity of angles from uniform distribution on the interval $[-\pi/2, \pi/2)$ for each sequence. For translation, we randomly sampled the start point and end point in the range of [-10, 10], and then moved the digit images on a straight line between the sampled points. We also experimented on sequential MNIST with background (Sequential MNIST-bg). For Sequantial MNIST-bg, we used the same generation rule as Sequential MNIST but we added background images behind the moving digits. For the background, we used a randomly sampled images from ImageNet [57], which were all resized to 32×32. Also, we only used the images of digit 4 for most of the experiments on Sequential MNIST/MNIST-bg. Unless otherwise noted, all evaluations in this paper for the Sequential MNIST are based on training with only *digit 4*.

3DShapes and SmallNORB are datasets with multiple factors of variation. We created a set of *constant-velocity* sequences from these datasets by varying a fixed combination of factors for each sequence. That is, on these datasets, we chose our $\mathcal{G}$ to be the set of variations of factors, and sampled each $g$ as $\prod_i g_i^{\ell_i}$, where $g_i$ represents the increase of $i$-th factor by one unit and $\ell_i \in \mathbb{Z}$. Thus, the value of $\ell_i$ represents the velocity in the direction of the $i$-th factor on the grid. For 3DShapes, we chose *wall hue*, *floor hue*, *object hue*, *scale*, and *orientation* as the factors to vary. We varied *elevation* and *azimuth* for SmallNORB. For the split of each sequence into $(\mathbf{s}_c, \mathbf{s}_p)$, we set $T_c = 2$ and $T_p = 1$ on all of the constant velocity experiments.

---

[2]$V : \mathcal{G} \to \{1\}$ is a valid irreducible representation for any $\mathcal{G}$.

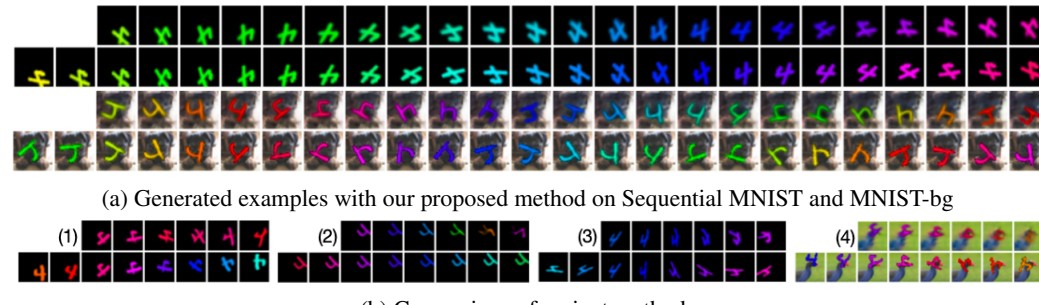

(a) Generated examples with our proposed method on Sequential MNIST and MNIST-bg

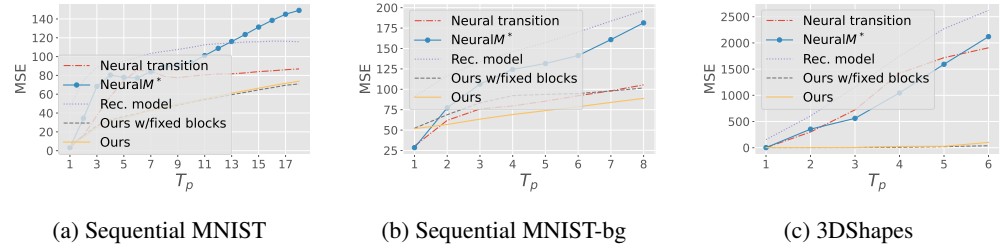

(b) Comparison of variant methods

Figure 4: (a): Predictions made by *meta-sequential prediction* on Sequential MNIST and MNIST-bg. The ground truth sequence is placed below the predictions, with the first two images representing $s_c$. (b): Typical failure examples generated by the comparative methods. (1)(2)(3) and (4) are Neural$M^*$, Neural transition, Rec. model and Ours w/ fixed block, respectively. See Appendix B for more examples.

(a) Sequential MNIST      (b) Sequential MNIST-bg      (c) 3DShapes

Figure 5: Prediction errors $\mathcal{L}^p$ with $T_c = 2$ and $T_p = 1, \ldots, 18$. During the training phase, models are trained to predict the observations only at $T_p = 1$. The prediction errors at $T_p > 1$ indicate the extrapolation performance. The results on SmallNORB can be found in Appnedix A.

We also conducted experiments for the sequences with constant acceleration on Sequential MNIST. To create a sequence with constant acceleration, we chose a pair $g_a, g_v \in \mathcal{G}$ for each sequence, and generated $s$ by setting $s_{t+1} = g_a^t g_v s_t$. We elaborate on the detail of this extension in 4.4.

As ablations, we tested several variants of our method: **fixed 2x2 blocks** (abbreviated as fixed blocks), **Neural$M^*$**, **Reconstruction model** (abbreviated as Rec. Model), and **Neural transition**. For the method of *fixed 2x2 blocks*, we separated the latent tensor $\Phi(s) \in \mathbb{R}^{16 \times 256}$ into 8 subtensors $\{\Phi^{(k)}(s) \in \mathbb{R}^{2 \times 256}\}_{k=1}^8$ and calculated pseudo inverse for each $k$ to compute the transition in each $\mathbb{R}^{2 \times 256}$ dimensional space. This variant yields $M^*$ as a direct sum of eight $2 \times 2$ matrices. We tested this variant to see the effect of introducing a predetermined representation theoretic structure as in [10]. For Rec. model, we trained $\Phi$ and $\Psi$ based on $\mathcal{L}^r$ in Eq. 3 with $T_c = 3$. We tested this variant to see the effect of our use of $T_p$. For Neural$M^*$, we trained an additional network $M_\theta$ that maps $s_c$ to a transition matrix, replaced $M^*$ with $M_\theta$ in (4), and optimized $\theta$ and $(\Phi, \Psi)$ simultaneously. We may see Neural$M^*$ as a variant in which the *meta* part of the internal and external training is removed from our method. For Neural transition, we trained 1x1 1D-convolutional networks to be applied to latent sequences *in the past* to produce the latent tensor in the next time step; for instance, $\tilde{s}_{t+1} = \Psi(1\text{DCNN}(\Phi(s_t), \Phi(s_{t-1})))$ when $T_c = 2$ [3]. The 1DCNN was applied along the multiplicity dimension $m$. In this variant, the relation between $\Phi(s_t)$ and $\Phi(s_{t+1})$ is not necessarily linear. Section D.1 in Appendix describes each of the comparison methods more in detail. In testing all of these variants, we used the same pair of encoder and decoder architecture as the proposed method.

## 4.1 Qualitative and quantitative results on the prediction

Figure 4 shows the example sequences generated by the proposed model and comparative models. Figure 5 presents the prediction performance at $T_c + T_p$ when $T_c = 2$. To produce this result, we

---

[3]This model can be seen as a simplified version of [60]

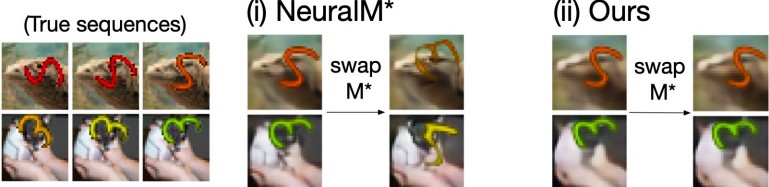

(a) Dependency of $M^*$ on sequence. The 2x3 tiled images in the leftmost panel represents two sequences $\mathbf{s}^{(1)}, \mathbf{s}^{(2)}$ with the same transition action $g$. We consider the effects of $M^{*(1)}$ and $M^{*(2)}$ inferred respectively from $\mathbf{s}^{(1)}$ and $\mathbf{s}^{(2)}$. Neural$M^*$ fails in prediction when $M^{*(2)}$ is used to predict $\mathbf{s}^{(1)}$. Our method does not fail by this swap, indicating $M^{*(2)} \cong M^{*(1)}$.

| Method | MNIST $\mathcal{L}^p$ | MNIST $\mathcal{L}^p_{\text{equiv}}$ | MNIST-bg $\mathcal{L}^p$ | MNIST-bg $\mathcal{L}^p_{\text{equiv}}$ | 3DShapes $\mathcal{L}^p$ | 3DShapes $\mathcal{L}^p_{\text{equiv}}$ | SmallNORB $\mathcal{L}^p$ | SmallNORB $\mathcal{L}^p_{\text{equiv}}$ |
|---|---|---|---|---|---|---|---|---|
| Rec. Model | 48.91 | 64.22 | 87.05 | 95.66 | 153.39 | 258.20 | 57.01 | 78.13 |
| Neural$M^*$ | 4.99 | 64.25 | 20.60 | 83.18 | 2.09 | 217.73 | 28.98 | 53.24 |
| MSP (Ours) | 6.42 | **15.91** | 27.38 | **36.41** | 2.74 | **2.87** | 31.14 | **44.77** |

(b) Equivariance performance based on $\mathcal{L}^p$(Eq.(2)) and $\mathcal{L}^p_{\text{equiv}}$(Eq.(6)) with $T_c = 2$ and $T_p = 1$. To evaluate equivariance errors on more difficult settings, we used all of digits in Sequential MNIST-bg for both training and test sets.

Figure 6: Quantitative and qualitative evaluation of learned equivariance.

back-propagated the prediction error at $T_p = 1$ to the encoder during the training, and the prediction at $T_p > 1$ was used to evaluate the extrapolation performance. Our method successfully predicts the images for $T_p \geq 1$. Neural transition and Neural$M^*$ had almost the same prediction performance at $T_p = 1$, but they both failed in extrapolation. Our *fixed 2×2 blocks* variant failed in extrapolation as well. This might be because the over-regularized structure of 2x2 block hindered with the training of the SGD optimization [17].

To evaluate how our learned representation relates to the structural features in the dataset, we also regressed the factors of transition from $M^*$ and regressed the class of the digits from $\Phi(s_1)$ (Figures 10 and 11 in Appendix A). SimCLR [8] and contrastive predictive coding (CPC) [61, 27] are tested as baselines. Please see Appendix D for the detailed experimental settings for SimCLR and CPC. Our method yields the representation with better prediction performance than the comparative methods on the test datasets.

## 4.2 Equivariance performance

As we have described through Section 3.1 and 3.2, the equivariance is achieved when $M^*(\mathbf{s}(s_1, g)|\Phi)$ in (2) does not depend on $s_1$, where we recall that $\mathbf{s}(s_1, g)$ represents the sequence that begins with $s_1$ and transitions with $g$. To see how much the trained model is equivariant to the transformations in the sequential dataset, we therefore calculated the *equivariance error*, which is the prediction error from applying $M^*(\mathbf{s}|\Phi)$ to $\Phi(s'_{T_c})$ for a pair $\mathbf{s} \neq \mathbf{s}'$ that transitions with the same $g$. In other words, when $T_c = 2$, we compute the following;

$$\mathcal{L}^p_{\text{equiv}} := \mathbb{E}_g \mathbb{E}_{\mathbf{s}, \mathbf{s}' \in \mathcal{S}(g)}[\|\Psi(M^*(\mathbf{s}|\Phi)\, \Phi(s'_2)) - s'_3\|^2_2] \tag{6}$$

where $\mathcal{S}(g)$ represents the set of all sequences that transition with $g$. For each pair of $\mathbf{s} \neq \mathbf{s}'$ we set $T_c = 2$ and $T_p = 1$ as done in the experiments in the previous sections. Table 6b compares the Neural$M^*$ method against our method in terms of the equivariance error. Figure 6a shows the result of applying $M^*(\mathbf{s}|\Phi)$ on $\Phi(s'_2)$ and applying $M^*(\mathbf{s}'|\Phi)$ on $\Phi(s_2)$. We see that when we swap $M^*$ this way, Neural$M^*$ also swaps the digits; this implies that $M^*$ learned by Neural$M^*$ encodes the *sequence* specific information together with the transition. On the other hand, swapping of $M^*$ does not affect the prediction for our method, suggesting that our method is succeeding to learn an equivariant model. This is somewhat surprising, because our model does not have the explicit mechanism to enforce the full equivariance ($P = I$ in Section 3.1).

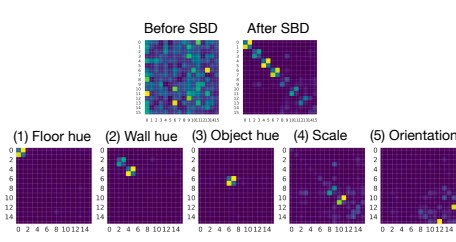

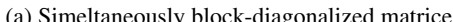

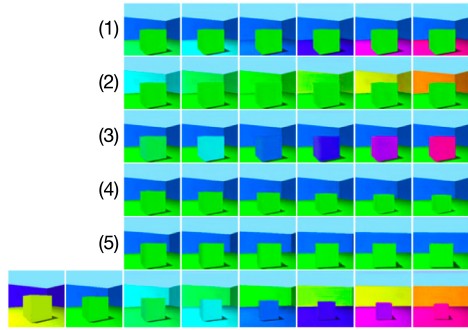

(a) Simeltaneously block-diagonalized matrices

(b) Disentangled transition representation on 3DShapes

Figure 7: (a) Simultaneous block-diagonalization (SBD) of $M^*$. The top right matrix is the visualization of $abs(V^* - I)$ averaged over all of the training sequences. Each of the five matrices below is the visualization of $abs(V^* - I)$ averaged over the set of sequences on which only a single factor was varied. Coordinates are permuted for better visibility. (b) Sequences generated by applying the transformation of just one block. To produce the disentangled sequences in each row from the leftmost two images in the bottom row, we performed the internal optimization of $M^*$ while setting *all but the block positions corresponding to each factor of variation* to be identity. We elaborate this result and the results for Sequential MNIST, MNIST-bg and SmallNORB in Appendix A.

### 4.3 Structures found by simultaneous block-diagonalization of $M^*$s

We have seen in the previous section that the trained $\Phi$ is fully equivariant to transformations $\mathcal{G}$, which implies each $M^*$ is a representation of the corresponding transformation of $g \in \mathcal{G}$. As we describe in Section 3.3, we apply simultaneous block-diaognalization to uncover *the symmetry structure* captured by $M^*$s. Figure 7a shows the structure revealed by simultaneous block-diagonalization through the change of basis $U$ trained by minimizing the average of $\mathcal{L}_{\mathrm{bd}}$ in eq.(5) over all $\mathbf{s}$. Figure 7b shows the results of applying transformation of only one block. We can see that each block only alters one factor of variation. Our results suggest that the learned $M^*$ captures the hidden disentangled structure of the group actions behind the datasets.

### 4.4 Extension to the sequences with constant acceleration

We have seen that *meta-sequential prediction* successfully learns an equivariant structure from the set of constant-velocity sequences. In this section, we show that we can extend our concept to the set of sequences sharing the stationarity of higher order (constant acceleration). By definition, the pair of $\Phi(s_t)$ and $\Phi(s_{t-1})$ encodes the information about the velocity at $t$. When the multiplicity $m$ is sufficiently large[4], the velocity can be estimated by: $^1M_t = \Phi(s_t)\Phi(s_{t-1})^\dagger$. Because this would yield a sequence of velocities, we can simply apply our method again to estimate the constant acceleration by $^2M^* = {}^1M_{+1}{}^1M_{+0}^\dagger$ where $^1M_{+0} = [{}^1M_2; ...; {}^1M_{T_c-1}] \in \mathbb{R}^{a \times (T_c-2)a}$ $^1M_{+1} = [{}^1M_3; ...; {}^1M_{T_c}] \in \mathbb{R}^{a \times (T_c-2)a}$. We can then predict the future representation $\tilde{s}_t$ for $t = T_c + 1, ..., T_p$ by

$$\tilde{s}_t = \Psi\left(\left(\prod_{t'=T_c+1}^{t} {}^1\tilde{M}_{t'}\right)\Phi(s_{T_c})\right) \text{ where } {}^1\tilde{M}_{t'} = {}^2M^{*(t'-T_c)} \, {}^1M_{T_c}. \quad (7)$$

where $\prod$ represents multiplications from left. We train $\Phi$ and $\Psi$ by minimizing the mean squared error between $\tilde{s}_t$ and $s_t$ for $t = T_c + 1, ..., T$ as in Eq.(2). To create a sequence of constant acceleration from MNIST dataset, we only used shape and color rotations. We chose the initial velocity for these rotations randomly on the interval $[-\pi/5, \pi/5]$ for each sequence, and chose the acceleration on the interval $[-\pi/40, \pi/40)$. The results are shown in Figure 8. The Neural transition and the constant-velocity version of our method failed to predict the accelerated sequence, while the 2nd order model succeeded in predicting the sequence even after $T_p > 5$. Also, Figure 15f and Table 16c

---

[4]If $m$ is less than $a$, we cannot obtain the pseudo inverse because of the rank deficient in $\Phi(s_{t-1})\Phi(s_{t-1})^\mathrm{T}$. Thus $m$ should be at least larger than $a$.

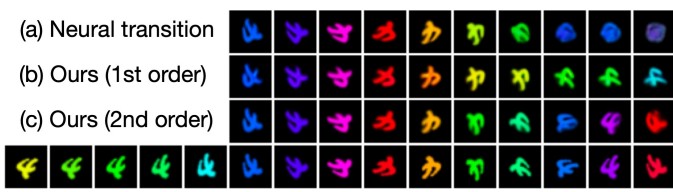

(a) Neural transition

(b) Ours (1st order)

(c) Ours (2nd order)

(a) Generated images on *accelerated* Sequential MNIST.

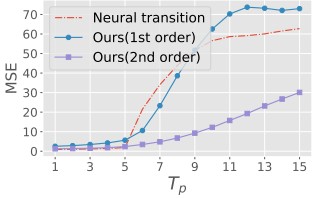

(b) Prediction error.

Figure 8: Results on accelerated Sequential MNIST. Every model was trained with $T_c = 5$, $T_p = 5$. Neural transition overfitted and collapsed from $T_p = 5$ (beyond the training horizon).

in Appendix shows that the accelerated version of our proposed model again achieves learning the equivariance relation.

## 5 Discussion & Limitations

**How is full equivariance achieved in our method?** The theoretical results we provided in section 3.1 only assure that $M(g, x)$ and $M(g, x')$ are similar when the underlying group is commutative, compact and connected. However, as we have shown experimentally, our method seems to be learning $\Phi$ for which the estimators of $M$ satisfy $M^*(\mathbf{s}(g, x)|\Phi) \cong M^*(\mathbf{s}(g, x')|\Phi)$. This can be happening because our framework and the training method based on the internal optimization in the latent space is somehow encouraging $M^*$ to be orthogonal (See the loss curve of orthogonality of $M^*$ in Appendix A). Maybe this is forcing the change of matrix $P$ such that $PM(g, x)P^{-1} = M(g, x')$ to be also rotations as well, which commutes with $M(g, x)$ itself. Also, Figure 4b and 5 show that, as reported in [34], the models trained with reconstruction loss like (3) does not well capture the group transformation behind the sequences: the encoder representation was found to be significantly worse than that of the model trained with (2). We hypothesize that (3) fails to remove the sequence-specific information from $M^*$, while (3) succeeds to do so by training the model to be able to predict the unseen images.

**Towards learning symmetries from more realistic observations** As we are making a connection between our prediction framework and group equivariance, we are essentially assuming that the transitions are always invertible, because group is closed under inversion. However, this might not be always the case in real world applications; for instance, if the image sequences are the sequential renderings of a rotating 3D object, the transitions are generally not invertible because only a part of the object is visible at each time step. We experimented Sequential ShapeNet, which is created from ShapeNet [7] dataset. A series of rendered images is generated by sequentially applying 3D rotations of different speeds for each axis. Generated results on Sequential ShapeNet (See 26 in Appendix B show that actually our current method was not able to generate the images on 3D rotated datasets. If the transitions are not invertible, some measures must be taken in order to resolve the indeterminacy, such as probabilistic modeling or additional structural inductive bias.

**Broader impact** Because our study generally contributes to predictions and extrapolation, it has as much potential to negatively affect the society as most other prediction methods. In particular, applications of our method to image sequence can be potentially integrated into weapon systems, for example. At the same time, our unsupervised learning of the symmetrical structure from sequential datasets may also contribute to new discoveries in the systems of finance, medical science, physics and other fields of ML such as reinforcement learning.

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
