# Appendix

## Table of Contents

## A  Supplemental results

### A.1  Qualitative and quantitative results on the prediction

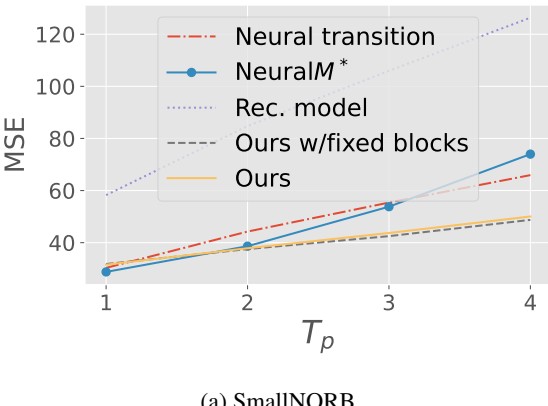

(a) SmallNORB

Figure 9: Prediction errors on smallNORB. During the training phase, the models were trained to predict the observations only at $t_{\mathrm{p}} = 1$. The prediction errors at $t_{\mathrm{p}} > 1$ indicate the extrapolation performance.

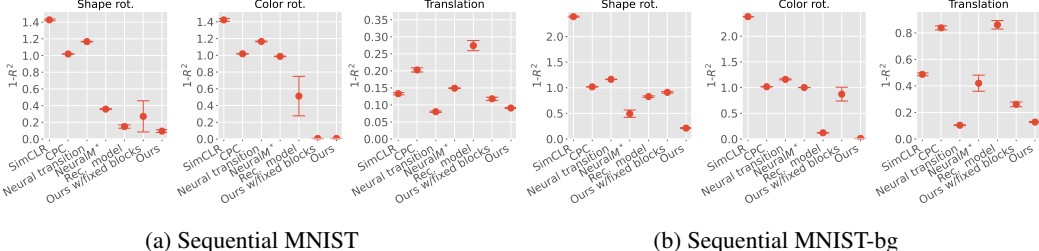

(a) Sequential MNIST                    (b) Sequential MNIST-bg

Figure 10: The results of linearly regressing the true transition parameters from $M^*$. For the performance evaluation, we used $1 - R^2$ scores (The value of 0 indicates the perfect prediction and 1 indicates the performance is chance level. $1 - R^2 > 1$ can happen when the model significantly overfits to the training set). For the color rotation and the shape rotation, $(\cos(v), \sin(v))$ was used as the target value where $v$ is the angle velocity. For this experiment, we trained the models on a set of sequences generated from *digit 4 class* only, and trained/evaluated the linear regression performance on the trained models' features on a set of sequences created from all digit classes in MNIST. Because SimCLR, CPC and Neural transition do not directly compute $M^*$, the linear regression was computed from the concatenation of the two consecutive latent representations that were used in our method for the computation of $M^*$.

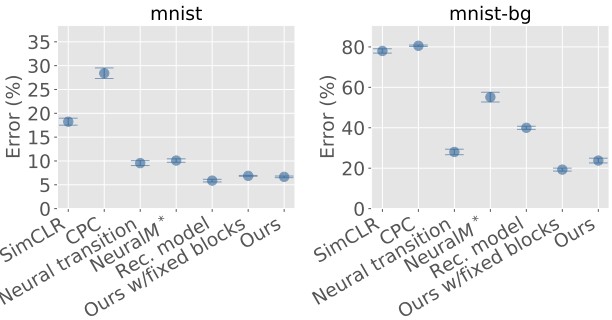

Figure 11: The results of digit classification evaluation on the sequential MNIST and MNIST-bg datasets. For this experiment, we trained the models on a set of sequences generated from only *digit 4 class*. We trained and evaluated the softmax classifier on the feature $\Phi(s_1)$ where $s_1$s are generated from all digit classes in MNIST.

## A.2 Equivariance performance

Figure 12 shows $(M^*(\mathbf{s}) - M^*(\mathbf{s}'))^2$ for the pairs of sequences that transition with same $g$ (e.g. $\mathbf{s} = \mathbf{s}(s_1, g), \mathbf{s}' = \mathbf{s}'(s_1', g)$). We see that $M^*$s computed from the representation learned by our method do not differ across $\mathbf{s}$ and $\mathbf{s}'$. This can also be confirmed visually in the generated sequences as well (Figure 13).

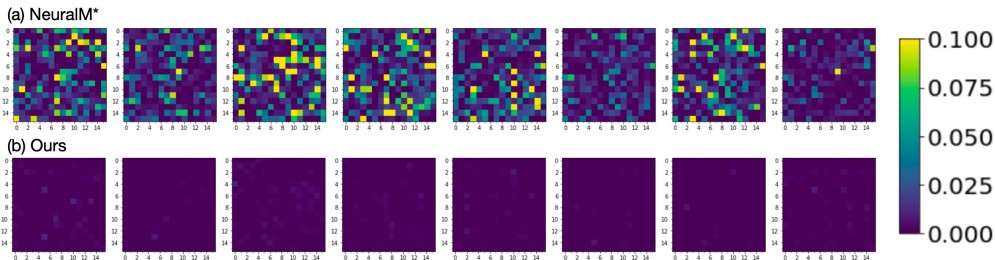

Figure 12: Visualization of $(M^*(\mathbf{s}) - M^*(\mathbf{s}'))^2$ where $\mathbf{s}, \mathbf{s}'$ that transition with the same $g$.

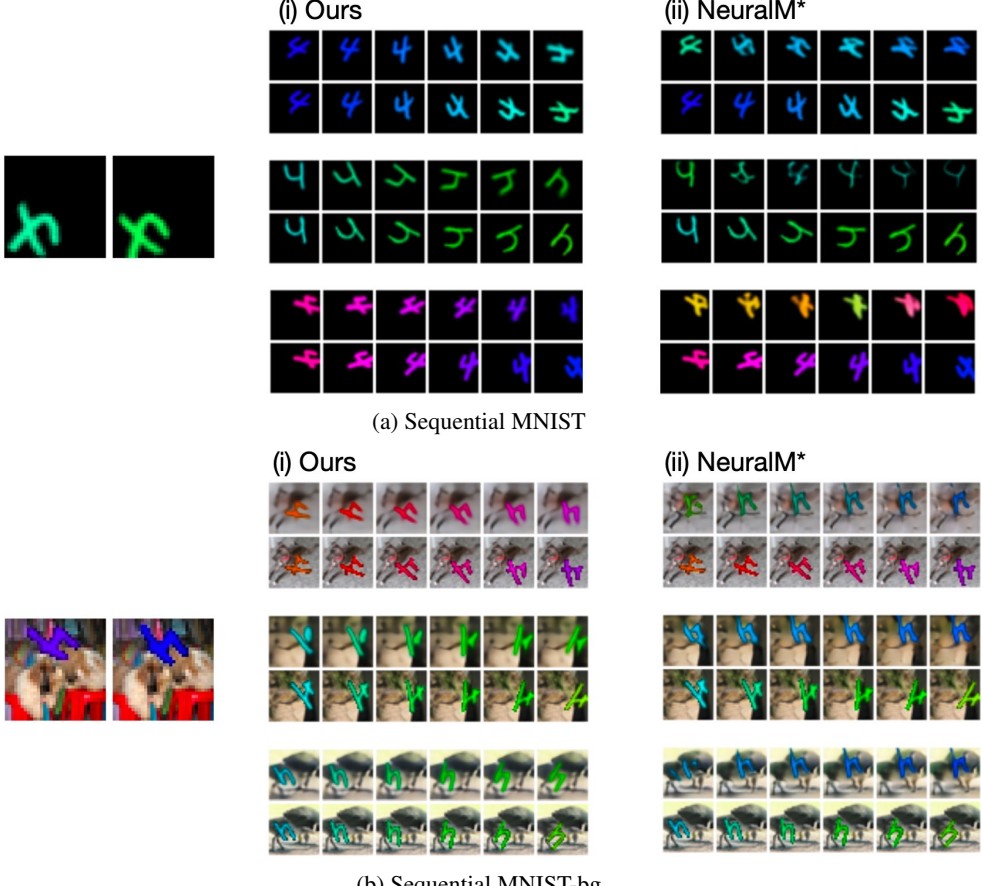

Figure 13: The result of transferring $M^*$ computed from one sequence to other sequences. For both sequential MNIST and sequential MNIST-bg, $M^*$ was computed from the two consecutive images placed on the left edge of the figure. In each pair of rows shown on the right, the top row corresponds to the generated sequence and the bottom row corresponds to the ground truth sequence that transitions with the same $g$ that was used to create the two consecutive images on the left. We see that each $M^*$ computed from our representation acts on different sequences in the same way.

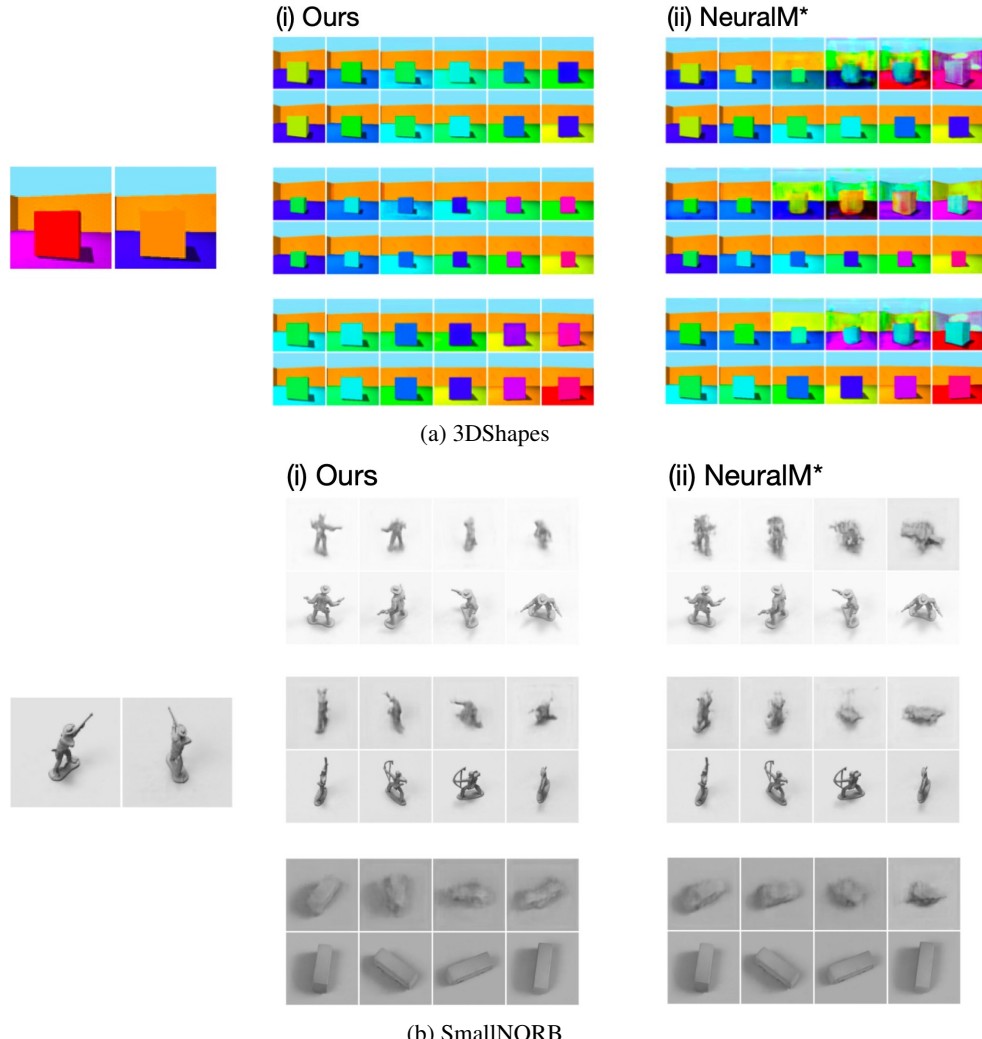

(a) 3DShapes

(b) SmallNORB

Figure 14: The result of transferring $M^*$ on 3DShapes and SmallNORB. The visualization follows the same protocol as in Figure 13.

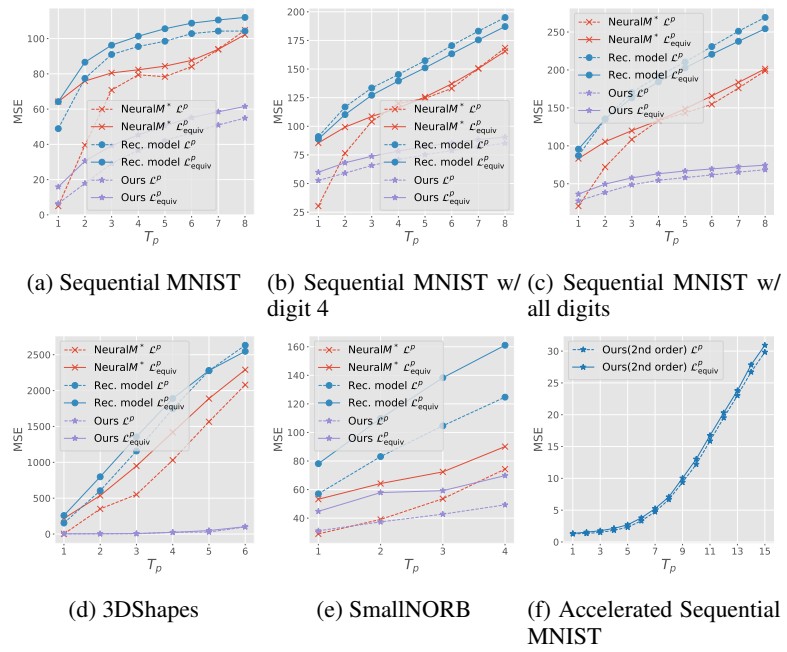

| (a) Sequential MNIST | (b) Sequential MNIST w/ digit 4 | (c) Sequential MNIST w/ all digits |
|---|---|---|

| (d) 3DShapes | (e) SmallNORB | (f) Accelerated Sequential MNIST |
|---|---|---|

Figure 15: Comparison of the prediction errrors and equivariance errors at $T_\mathrm{p} \geq 1$.

|  | Seq. MNIST | | Seq. MNIST-bg (w/ digit 4) | |
|---|---|---|---|---|
| Method | $\mathcal{L}^p$ | $\mathcal{L}^p_{\text{equiv}}$ | $\mathcal{L}^p$ | $\mathcal{L}^p_{\text{equiv}}$ |
| Rec. Model | 48.91±4.47 | 64.22±5.69 | 91.02±2.22 | 88.93±2.89 |
| Neural$M^*$ | 4.99±0.87 | 64.25±2.59 | 30.32±0.36 | 85.46±2.66 |
| MSP (Ours) | 6.42±0.21 | **15.91**±0.49 | 52.67±0.86 | **59.87**±1.37 |

(a) Equivariance performance on sequential MNIST and MNIST-bg w/ digit 4

|  | Seq. MNIST-bg (w/ all digits) | | 3DShapes | |
|---|---|---|---|---|
| Method | $\mathcal{L}^p$ | $\mathcal{L}^p_{\text{equiv}}$ | $\mathcal{L}^p$ | $\mathcal{L}^p_{\text{equiv}}$ |
| Rec. Model | 87.05±3.32 | 95.66±7.71 | 153.39±24.1 | 258.20±25.8 |
| Neural$M^*$ | 20.60±0.25 | 83.18±2.50 | 2.09±0.12 | 217.73±46.7 |
| MSP (Ours) | 27.38±0.14 | **36.42**±0.08 | 2.75±0.25 | **2.87**±0.30 |

(b) Equivariance performance on sequential MNIST-bg w/ all digits and 3DShapes

|  | SmallNORB | | Accelerated Seq. MNIST | |
|---|---|---|---|---|
| Method | $\mathcal{L}^p$ | $\mathcal{L}^p_{\text{equiv}}$ | $\mathcal{L}^p$ | $\mathcal{L}^p_{\text{equiv}}$ |
| Rec. Model | 57.01±2.69 | 78.14±4.42 | | |
| Neural$M^*$ | 28.98±1.25 | 53.24±0.64 | | |
| MSP (Ours) | 31.14±0.52 | **44.77**±0.38 | 1.27± 0.02 | 1.34 ± 0.03 |

(c) Equivariance performance on SmallNORB and accelerated sequential MNIST

Figure 16: More detailed version of Fig 15 with standard deviation values. The statistics in this figure were calculated over three models initialized with different random seeds. For the definition of $\mathcal{L}^p$ and $\mathcal{L}^p_{\text{equiv}}$, see (2) and (6).

## A.3 More results on simultaneous block-diagonalization

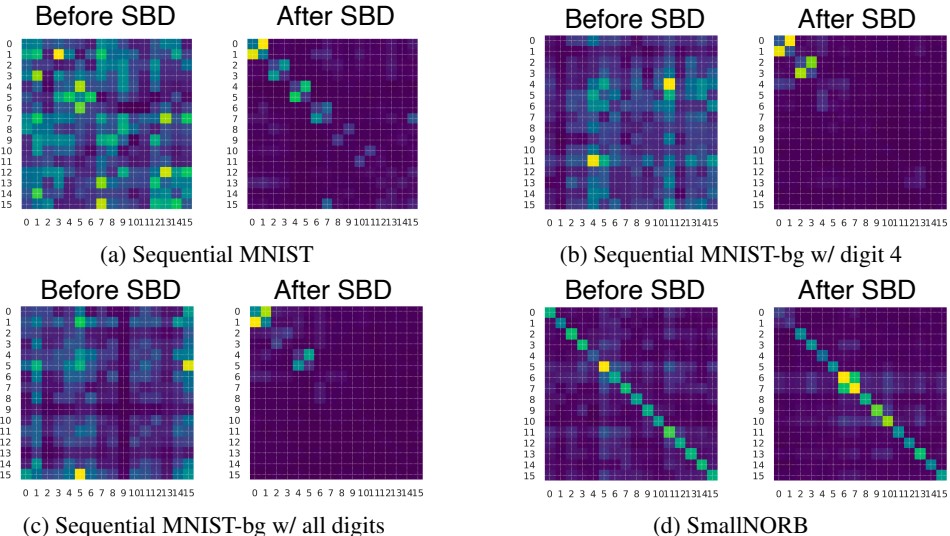

(a) Sequential MNIST

(b) Sequential MNIST-bg w/ digit 4

(c) Sequential MNIST-bg w/ all digits

(d) SmallNORB

Figure 17: The visualization of simultaneously block-diagonalized (SBD) matrices for Sequential MNIST/MNIST-bg and SmallNORB datasets. As in Figure 7a, our visualizations correspond to $abs(M^* - I)$ and $abs(V^* - I)$ instead of the raw matrices ($V^*$ is the block-diagonalized version of $M^*$. See Section 3.3 and Section E).

Figure 17 is the visualization of the block structures revealed by the simultaneous block-diagonalization on Sequential MNIST/MNIST-bg and SmallNORB. The detail of the block-diagonalization method is provided in Section 3.3 and Section E.

To investigate what type of transformations these blocks correspond to, we studied the effect of using just one particular set of blocks in the block diagonalized transition matrix (Figure 18). To create the transformation of *one particular set of blocks*, we modified the block-diagonalized $M^*$ by setting all block positions other than the target blocks to identity. We can visually confirm that disentanglement is achieved by the partition of block positions. See the figure captions for more details.

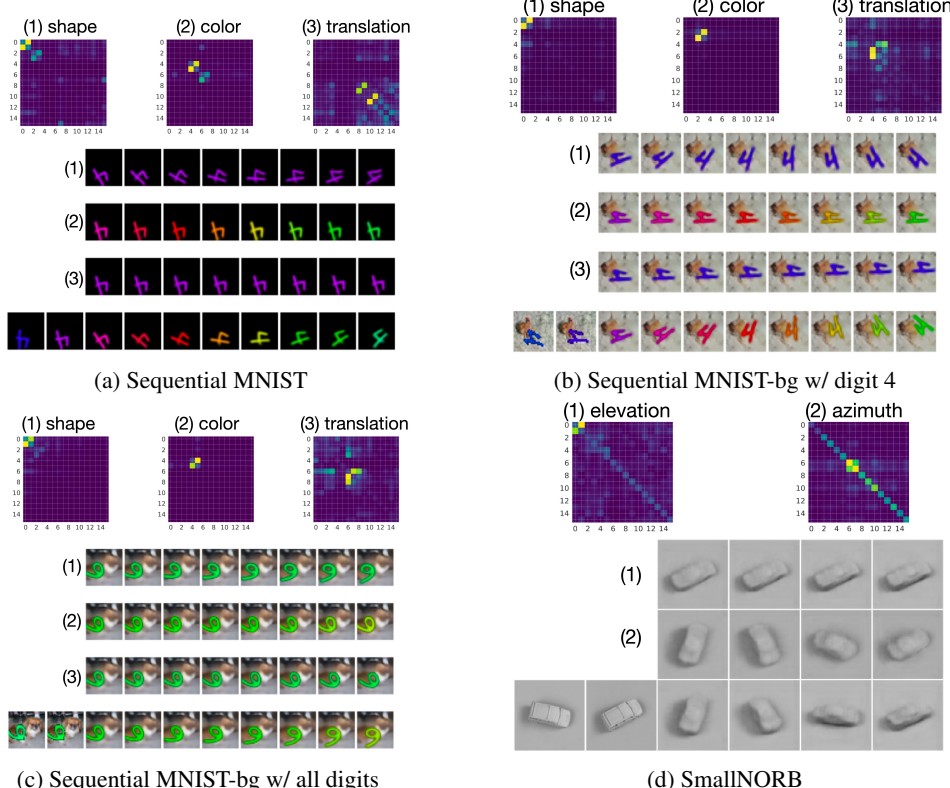

(a) Sequential MNIST

(b) Sequential MNIST-bg w/ digit 4

(c) Sequential MNIST-bg w/ all digits

(d) SmallNORB

Figure 18: Generation of disentangled sequences. The bottom sequence in each frame of this figure is the ground truth. We generated each one of (1), (2) (and (3)) by applying the transformation corresponding to only one particular set of the blocks. To create each sequence, we first computed $M^*$ from the first two time-steps($t = 1, t = 2$) in the ground truth, and block-diagonalized $\bar{M}^*$ to obtain $\hat{M}^*$. We then created the transformation corresponding to only *one particular set of blocks* by setting all the block positions of $\hat{M}^*$ other than the target blocks to identity. We then applied the powers of the one-block-set transformation to the image at $t = 2$ to generate the disentangled future sequence. The assignment of block positions to disentangled factors was found manually by looking at the activated blocks when we altered one factor in the ground truth sequences. See Table 1 for the correspondence between block positions and disentangled factors. We can visually confirm that disentanglement is achieved through block partitions.

| dataset | The *factor-block position* correspondence |
|---|---|
| Sequential MNIST | (1) $\{0, 1, 2, 3\}$, (2) $\{4, 5, 6, 7\}$, (2) $\{8, 9, 10, 11\}$ |
| Sequential MNIST-bg w/ digit 4 | (1) $\{0, 1\}$, (2) $\{2, 3\}$, (3) $\{4, 5, 6, 7, 8\}$ |
| Sequential MNIST-bg w/ all digits | (1) $\{0, 1, 2, 3\}$, (2) $\{4, 5\}$, (3) $\{6, 7, 8\}$, |
| 3DShapes | (1) $\{0, 1\}$, (2) $\{2, 3, 4, 5\}$, (3) $\{6, 7\}$, |
| | (4) $\{8, 9, 10, 11\}$, (5) $\{12, 13, 14, 15\}$ |
| SmallNORB | (1) $\{0, 1, 2, 3, 4, 5\}$, (2) $\{6, 7, 8, 9, 10, 11, 12, 13\}$ |

Table 1: The correspondence between block positions and disentangled factors in simultaneous block-diagonalization. For each $i$, " $(i)\{a_1, a_2, ...a_m\}$" means that the $i$-th disentangled factor has coordinates $\{a_1, a_2, ...a_m\}$. For example, the block that is positioned at coordinates $\{4, 5\}$ changes the second disentangled factor (shape rotation) in Sequential MNIST.

## A.4 Orthgonality of $M^*$ during training

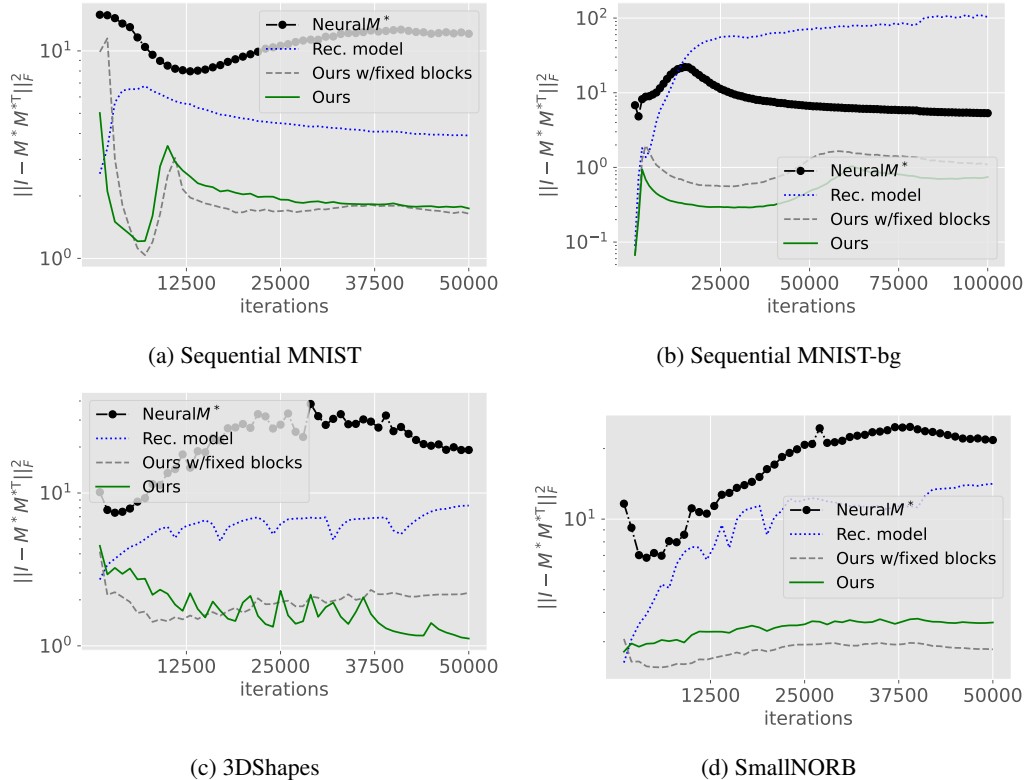

(a) Sequential MNIST

(b) Sequential MNIST-bg

(c) 3DShapes

(d) SmallNORB

Figure 19: The transition of $\|I - M^* M^{*\mathrm{T}}\|_F^2$ during the training. We can observe that, for our method, the learned representation evolves in such a way that the estimated transition $M^*$ tends to become orthogonal.

## B Generated examples

Figures 20-26 show the seqeuences generated by our method and its variants for each dataset. The visualization follows the same protocol as in Figure 4.

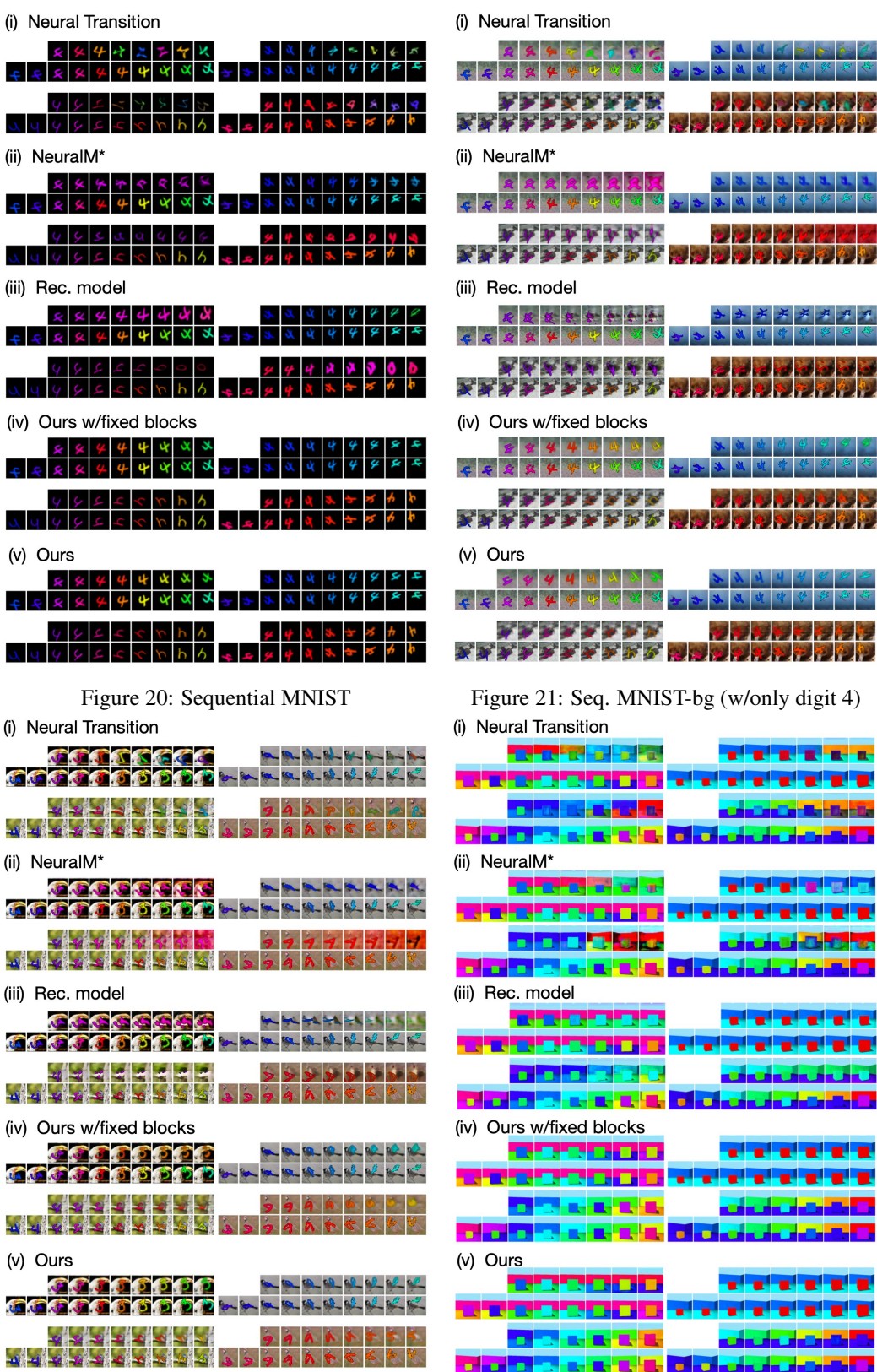

Figure 20: Sequential MNIST

Figure 21: Seq. MNIST-bg (w/only digit 4)

Figure 22: Seq. MNIST-bg (w/all digits)

Figure 23: 3DShapes

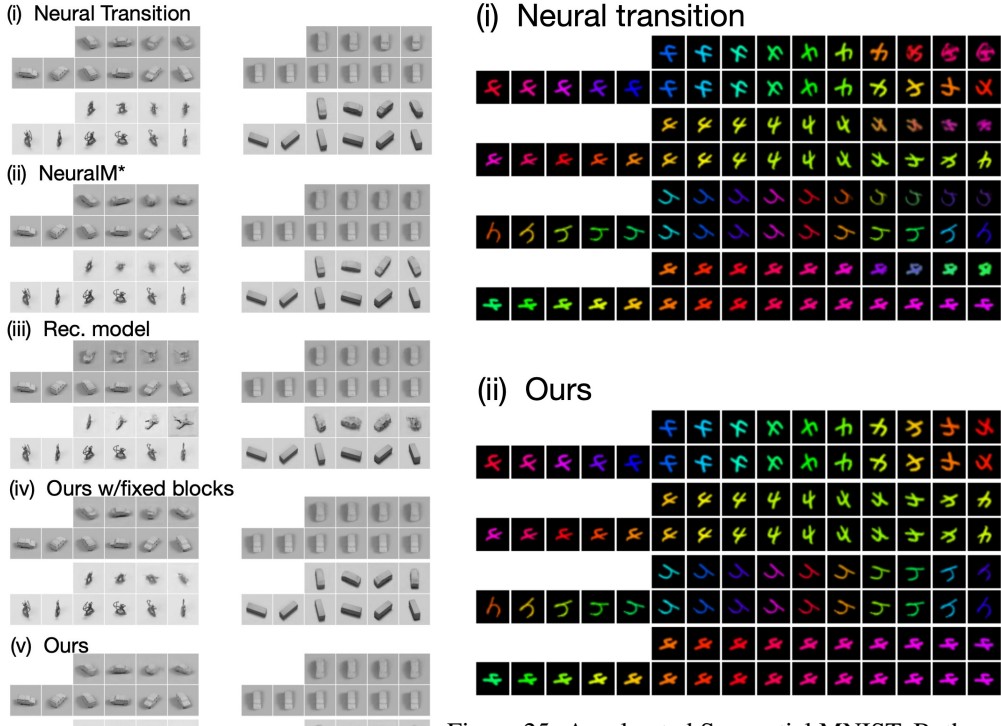

(i) Neural Transition

(ii) NeuralM*

(iii) Rec. model

(iv) Ours w/fixed blocks

(v) Ours

Figure 24: SmallNORB

(i) Neural transition

(ii) Ours

Figure 25: Accelerated Sequential MNIST. Both models were trained with $T_c = 5$ and $T_p = 5$. For the training procedure on this experiment, please see Section 4.4.

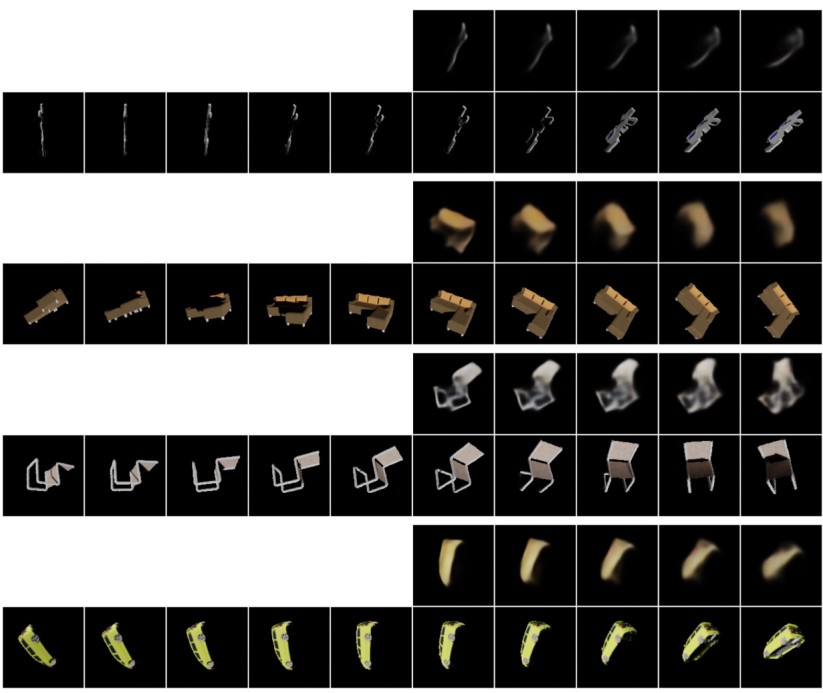

Figure 26: Sequential ShapeNet generated by the proposed method. The model was trained with $T_c = 5$ and $T_p = 5$. Our method cannot make good predictions on this dataset. Note that, unlike other datasets we studied on this paper, the transition in Sequential ShapeNet is not necessarily invertible, because some parts of a $3D$ object are often not visible in the 2D rendering.

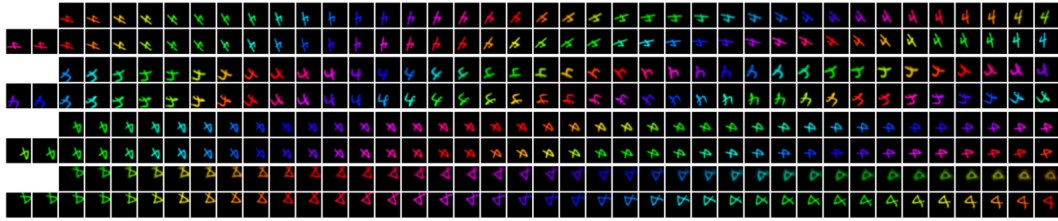

Figure 27: Extrapolation results for longer future horizon ($t_\mathrm{p} = 1,...,38$). The visualization follows the same protocol as in Figure 4a.

## C  Algorithm

We provide the algorithmic description of our method. Definitions of all symbols in the table are the same as in the main sections.

---

**Algorithm 1** Calculate the loss over $\Phi$ and $\Psi$.

---

Input: Given an encoder $\Phi$, a decoder $\Psi$ and a sequence of observations $\mathbf{s} = [s_1, \ldots, s_{T_c}, s_{T_c+1}, \ldots, s_T]$.

1. Encode the observations into latent variables $H_t = \Phi(s_t)$ for $s_1, ..., s_{T_c}$.
2. Estimate the transition matrix by solving linear problem: $M^* = H_{+1} H_{+0}^\dagger$ where $H_{+0}$ and $H_{+1}$ are the horizontal concatenation $[H_1; H_2; ...; H_{T_c-1}]$ and $[H_2; H_3; , ...,; H_{T_c}]$, respectively.
3. Predict the future sequence by : $\tilde{s}_t = \Psi((M^*)^{t-T_c} H_{T_c})$ for $t = T_c + 1, ..., T$.
4. Calculate the loss for the sequence $\mathbf{s}$: $\sum_{t=T_c+1}^{T} \|\tilde{s}_t - s_t\|_2^2$

---

## D  Experimental settings

### D.1  Ablation studies

As ablations, we tested several variants of our method: **fixed 2x2 blocks**, **Neural$M^*$**, **Reconstruction model** (abbreviated as Rec. Model), and **Neural transition**. We describe each one of them below.

- **Fixed 2x2 blocks**: For this model, we separated the latent tensor $\Phi(s) \in \mathbb{R}^{16 \times 256}$ into 8 subtensors $\{\Phi^{(k)}(s) \in \mathbb{R}^{2 \times 256}\}_{k=1}^8$ and calculated the pseudo inverse for each $k$ to compute the transition in each $\mathbb{R}^{2 \times m}$ dimensional space. Essentially, this variant of our proposed method computes $M^*$ as a direct sum of eight $2 \times 2$ matrices. In the pioneer work of [10] that endeavors to learn the symmetry in a linear system using the representation theory of commutative algebra, the authors hard-code the irreducible representations/block matrices in their model. Our study is distinctive from many applications of representation theory and symmetry learning in that we uncover the symmetry underlying the dataset not by introducing any explicit structure, but by simply seeking to improve the prediction performance. We therefore wanted to experiment how the introduction of the hard-coded symmetry like the one in [10] would affect the prediction performance.

- **Neural$M^*$**: Our method is "meta" in that we distinguish the internal training of $M^*$ for each sequence from the external training of the encoder $\Phi$. Put in another way, the internal optimization process of $M^*$ itself is the function of the encoder. To measure how important it is to train the encoder with such a meta approach, we evaluated the performance of Neural$M^*$ approach. To reiterate, Neural$M^*$ uses a neural network $M_\theta^*$ that directly outputs $M^*$ on the conditional sequence, and train the encoder and the decoder via

$$\sum_{t=T_c+1}^{T_c+T_p} \|\Psi(M_\theta^*(\mathbf{s}_c)^{t-T_c} \Phi(s_{T_c})) - s_t\|_2^2,$$

thereby testing the training framework that is similar to our method "minus" the "meta" component.

- **Reconstruction model (Rec. model)**: In our default algorithm, we train our encoder and decoder with the prediction loss $\mathcal{L}^p$ in eq.(2) over the future horizon of length $T_p - T_c$. We therefore wanted to verify what would happen to the learned representation when we train the model with the reconstruction loss $\mathcal{L}^r$ in eq.(3) in which the model predicts the observations contained in the conditional sequence. Specifically, we trained $\Phi$ and $\Psi$ based on $\mathcal{L}^r$ in (3) with $T = T_c = 3$.

- **Neural Transition**: One important inductive bias that we introduce in our model is that we assume the latent transition to be linear. We therefore wanted to test what happens to the results of our experiment if we drop this inductive bias. For Neural transition, we trained a network with 1x1 1D-convolutions that inputs $\Phi(s)$ in the past to produce the latent tensor in the next time step; for instance, $\tilde{H}_{t+1} = \text{1DCNN}(\Phi(s_t), \Phi(s_{t-1}))$ when $T_c = 2$. This model can be seen as a simplified version of [60]. The 1DCNN was applied along the multiplicity dimension ($m$).

In testing all of these variants, we used the same pair of encoder and decoder architecture as the proposed method.

## D.2 Training details

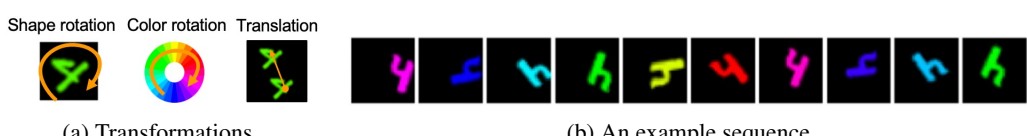

(a) Transformations           (b) An example sequence

Figure 28: *Sequential MNIST* dataset. The transition in each sequence was produced by combining three families of actions: shape rotation, color rotation and translation.

For the model optimization in every experiment, we used ADAM [37]. The number of iterations for the parameter updates were $50,000$ on Sequential MNIST, 3Dshapes, and SmallNORB. The number of iterations was $100,000$ on Sequential MNIST-bg (with only digit 4 class) and accelerated Sequential MNIST. For MNIST-bg (with all digits) and Sequential ShapeNet, the number of iterations was $200,000$. We set the initial learning rate of ADAM optimizer to $0.0003$ and decayed it to $0.0001$ after a certain number of iterations. For Sequential MNIST, 3Dshapes, and SmallNORB, we began the decay at $40,000$-th iteration. For Sequential MNIST-bg (with digit 4 only) and accelerated Sequential MNIST, we began the decay at $80,000$-th iteration. For Sequential MNIST-bg (with all digits) and Sequential ShapeNet, we began the decay at $160,000$-th iteration.

The batchsize was set to 32 for all experiments. We conducted all experiments on NVIDIA A100 GPUs. Training our proposed model takes approximately one hour per $50,000$ iterations on a single A100 GPU. Total amount of time to reproduce the full results in our experiments is approximately 12 days on a single A100 GPU.

We found that the choice of the latent dimension ($a \times m$) does not make significant difference on the results as long as they are not too small (For example, if $G$ is a torus group consisting of $n$ commuting axis, $a$ must be no less than $2n$ when all the observations are real-valued; otherwise the model will underfit the datasets. Also, we chose $m$ to be larger than $a$ so that $\Phi(x)$ becomes full rank almost surely. This allows us to solve $M^*$ from $M^*\Phi(s_1) = \Phi(s_2)$ (We can compute $M^*$ with $T_c = 2$.) Choosing $m > a$ also plays a role in the theory (Section F).

As for the Neural $M^*$ and Neural transition models, we also optimized the invertibility loss: $\sum_{t=1}^{T_c} \|\Psi(\Phi(s_t)) - s_t\|_2^2$ in addition to $\mathcal{L}^p$. Adding this loss to the original objective yielded better results for these models in terms of prediction error and equivariance error for all experiments.

**SimCLR and CPC settings in the downstream task experiments in Figure 10 and 11** To evaluate our method as a representation learning method, we compared our method against SimCLR [8] and CPC [61, 27]. For SimCLR, we treated any pair of observations in the same sequence as a positive pair, and any pair of observations in different sequences as a negative pair. We used the

same encoder architecture as in our baseline experiment for both SimCLR and CPC. For the projection head of SimCLR, however, we used the same architecture as in the original paper. For the auto regressive network of latent representation in CPC, we used the same architecture as in Neural transition (see Section 4). The latent dimension was set to 512 for both models. We experimented with larger and smaller dimensions as well, but however large the difference, altering the dimensions did not result in significant improvements in terms of the representation quality evaluated in the experimental sections. The temperature parameters for the logit output were searched in the range of [1e-3, 1e-2, 1e-1, 1.0, 10.0]. Because SimCLR is not built for the sequential dataset, it is not expected to perform too well in terms of regression performance. We however evaluated these models as minimum performance baselines.

### D.3    Additional details of datasets

Our training-test split was the same as the split in the original dataset. Therefore the train-test split of Sequential MNIST/MNIST-bg was the same as that of MNIST, and the split of the SmallNORB dataset we used was the same as that of the original SmallNORB. Meanwhile, the 3DShapes dataset does not have train-test split, so we conducted the training and the test evaluation on the same dataset for the sequential 3DShapes experiment. We also used only cubic shape examples on the 3DShapes experiments. For Sequential ShapeNet, 90% of objects in the original ShapeNetCore assets were used for the training and the rest were used for the evaluation. The input size of each example in a given sequence was $3 \times 32 \times 32$ for Sequential MNIST/MNIST-bg, $3 \times 64 \times 64$ for 3DShapes, $1 \times 96 \times 96$ for SmallNORB, and $3 \times 128 \times 128$ for Sequential ShapeNet.

To generate Sequential ShapeNet, we used Kubric [21] to render the objects in ShapeNet [7] datasets. For each sequence, we sampled one object from ShapeNetCore assets, and used 3D rotation to define the transition. The angle of 3D rotation in each axis(xyz) was sampled from the uniform distribution over the interval $[0, \pi/4)$.

### D.4    Network architecture

We used ResNet-based encoder and decoder[26]. We used ReLU function [53, 20, 52] for each activation function and group normalization [68] for the normalization layer. We used weight standarization [55] for all of filters in each convolutional network. Also, we used trainable positional embedding in each block of the decoder, which was initialized to the 2D version of sinusoidal positional embeddings [66]. We provide the details of the architecture in Table 2 and Figure 29.

For the Neural$M^*$ method, we used the same model in the table 2a except the input channel of the network was set to 6 (and 2 for SmallNORB) because this method uses a pair of images ($s_1$, $s_2$) as an input.

For the Neural transition model, we used a network with 1x1 1D convolutions to map $[H_t, \ldots, H_{t+t'}]$ to $H_{t+t'+1}$. The network architecture is the 1x1 1D convolutional version of the table 2a without downsampling. The number of ResBlocks was set to two. We also replaced all of the group normalization layers with layer normalization [2].

|  | #channels or #dims | Resampling | Spatial Resolution |
|---|---|---|---|
| 3x3 2DConv | 32*k | - | H×W |
| ResBlock | 64*k | Down | (H/2)×(W/2) |
| ResBlock | 128*k | Down | (H/4)×(W/4) |
| ResBlock | 256*k | Down | (H/8)×(W/8) |
| GroupNorm | 256*k | - | (H/8)×(W/8) |
| ReLU | 256*k | - | (H/8)×(W/8) |
| Flatten | 256*k*(H/8)*(W/8) | - | - |
| Linear | 16*256 | - | - |

(a) Encoder architecture

|  | #channels or #dims | Resampling | Spatial Resolution |
|---|---|---|---|
| Linear | 256*k*(H/8)*(W/8) | - | - |
| Reshape | 256*k | - | (H/8)×(W/8) |
| ResBlock | 128*k | Up | (H/4)×(W/2) |
| ResBlock | 64*k | Up | (H/2)×(W/4) |
| ResBlock | 32*k | Up | H×W |
| GroupNorm | 32*k | - | H×W |
| ReLU | 32*k | - | H×W |
| 3x3 2DConv | 3 (1 for SmallNORB) | - | H×W |

(b) Decoder architecture

Table 2: The detail of the encoder and decoder architecture used in our experiments. The columns of '#channels or #dims' and 'Spatial resolution' respectively represent the channels/dimensions and the spatial resolution at the end of each corresponding module. 'Resampling' column represents whether the corresponding layer performs upsampling (Up), downsampling (Down) or none of them (-). Please see Figure 29 for the detail of the ResBlock architecture. The value $k$ in the table was set to 1 for 3DShapes, SmallNORB and Sequential ShapeNet. The value $k$ was set to 2 for Sequential MNIST and accelerated Sequential MNIST, and 4 for Sequential MNIST-bg. For Small-NORB, we added one more downsampling ResBlock after the third ResBlock in the encoder and one more upsampling ResBlock before the first ResBlock in the decoder. For Sequential ShapeNet, we added two more downsampling ResBlock in the encoder and two more upsampling Resblock in the decoder.

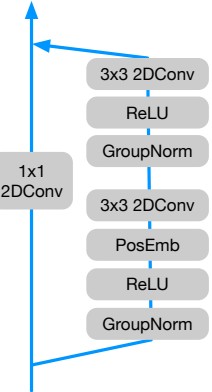

Figure 29: ResBlock architecture in the encoder and decoder. 'PosEmb' stands for the positional embedding layer which concatenates the learned positional embedding to its input. The embedding dimension was set to $32 \times H \times W$. The PosEmb layer was used only in decoder's resblock. For the encoder, we performed downsampling (mean average pooling) after the second convolution layer. For the decoder, we performed upsampling before the first convolution. Also, we added a downsampling layer (mean avegrage pooling) after the 1x1 convolution. For upsampling, we added a layer of nearest-neighbor upsampling before the 1x1 convolution. The number of groups for the group normalization layer was set to 32.

## E    Simultaneous Block-diagonalization

To find $U$ that simultaneously block-diagonalizes all $M^*(\mathbf{s}|\Phi)$, we optimized $U$ based on the objective function that measures the block-ness of $V^*(\mathbf{s}) := UM^*(\mathbf{s}|\Phi)U^{-1}$. Our objective function is based on the fact that, if we are given an adjacency matrix $A$ of a graph, then the number of connected components in the graph can be identified by looking at the rank of graph Laplacian:

$$\dim(\mathrm{Ker}(\Delta A)) = \#\text{of blocks in } A \tag{8}$$

where $\Delta$ is the graph Laplacian operator on $A$. To relate our $V^* := UM^*U^{-1}$ to a graph, we see it as a bipartite graph and calculate the adjacency matrix by:

$$A(V^*(\mathbf{s})) := abs(V^*(\mathbf{s}))abs(V^*(\mathbf{s}))^{\mathrm{T}} \tag{9}$$

where $abs(V^*(\mathbf{s}))$ represents the element-wise absolute value of $V^*$: $abs(M^*(\mathbf{s}))_{ij} = |M^*_{ij}|$. To optimize Eq.(8) with respect to the change of basis $U$ by continuous optimization, we used the lasso version of Eq.(8):

$$\mathcal{L}_{\mathrm{bd}}\left(V^*(\mathbf{s})\right) := \|\Delta\left(A\left(V^*(\mathbf{s})\right)\right)\|_{\mathrm{trace}} = \sum_{d=1}^{a} \sigma_d\left(A\left(V^*(\mathbf{s})\right)\right) \tag{10}$$

where $\sigma_i(\Delta A)$ is $i$-th singular value of $\Delta A$. We used the symmetrically normalized version of the graph Laplacian: $\Delta A = I - D^{-1/2}AD^{-1/2}$ where $D$ is the degree matrix of $A$. Summing this over all $\mathbf{s}^{(i)}$ in the dataset, we obtain: $\bar{\mathcal{L}}_{\mathrm{bd}} := \frac{1}{N}\sum_{i=1}^{N}\mathcal{L}_{\mathrm{bd}}\left(V^*(\mathbf{s}^{(i)})\right)$. We search for $U$ that simultaneously block-diagonalizes all $V^*(\mathbf{s}^{(i)})$ by minimizing $\bar{\mathcal{L}}_{\mathrm{bd}}$ w.r.t. $U$.

## F    Formal statements and the proofs of the theory section

We begin by summarizing the notations to be used in our formal statements. We use $\mathcal{X}$ to denote the space of all observations at a single time step, and $\Phi : \mathcal{X} \to \mathcal{H}$ to denote the encoder from $\mathcal{X}$ to the latent space $\mathcal{H}$. If $\mathbf{s} = [s_t \in \mathcal{X}; t = 1, ..., T]$ is one instance of video-sequence to be used in our training, we assume that, for each $\mathbf{s}$, there is an operator $g : \mathcal{X} \to \mathcal{X}$ such that $gs_t = s_{t+1}$ for each $t$. As such, each $\mathbf{s}$ is characterized by a pair of initial state $s_1 \in \mathcal{X}$ and $g \in \mathcal{G}$, where $\mathcal{G}$ is the set of all operators considered. Thus, we would use $\mathbf{s}(s_1, g)$ to denote a specific sequence.

Now, given a fixed encoder $\Phi : \mathcal{X} \to \mathbb{R}^{a \times m}$, our training process computes the transition matrix $M$ independently for each instance of $\mathbf{s}(s_1, g) = [s_t]_{t=1}^{T} = [g^{t-1}s_1]_{t=1}^{T}$. In particular, we compute

$$M^*(g, s_1|\Phi) = \arg\min_M \frac{1}{T}\sum_{t=1}^{T-1} \|\Phi(s_{t+1}) - M\Phi(s_t)\|_F^2. \tag{11}$$

In the theory developed here, we investigate the property of the optimal $\Phi$ when $T = \infty$ so that $M^*(g, s_1|\Phi)$ achieves

$$\|\Phi(s_{t+1}) - M^*(g, s_1|\Phi)\Phi(s_t)\|^2 = 0$$

or equivalently,

$$M^*(g, s_1|\Phi)\Phi(g^{t-1} \circ s_1) = \Phi(g^t \circ s_1)$$

for all $t \in \mathbb{N}$, $g \in \mathcal{G}$, and $s_1 \in \mathcal{X}$. We would like to know whether $M^*(g, x|\Phi)$ has no dependency on $x$ so that $M_g = M^*(g|\Phi)$ defines an equivariance relation and hence a group representation .

We begin tackling this problem by first investigating $M^*(g, x|\Phi)$ within an orbit $\mathcal{G} \circ x := \{h \circ x \in \mathcal{X} \mid h \in \mathcal{G}\}$. That is, we check if we can say $M^*(g, x|\Phi) = M^*(g, h \circ x|\Phi)$ for any $g, h \in \mathcal{G}$ and $x \in \mathcal{X}$. We call this property *intra-orbital homogeneity*.

We assume that $\mathcal{G}$ is a compact commutative Lie group in the following result.

**Proposition F.1** (Intra-orbital homogeneity). *Suppose that $\mathcal{G}$ is a compact commutative Lie group, $\Phi(x) \in \mathbb{R}^{a \times m}$ has rank $a$, and $M(g, x) \in \mathbb{R}^{a \times a}$ satisfies*

$$M(g, x)\Phi(g^k \circ x) = \Phi(g^{k+1} \circ x) \tag{12}$$

*for all $k \in \mathbb{N} \cup \{0\}$, $x \in \mathcal{X}$ and $g \in \mathcal{G}$. If $M(g, x)$ is continuous with respect to $g$ and is uniformly continuous with respect to $x$, then*

$$M(g, x) = M(g, h \circ x)$$

*for all $h \in \mathcal{G}$.*

Before going into the proof of this proposition, we show the following lemma about the basic properties of $M(g, x)$ that satisfies (12).

**Lemma F.2.** *Assume that $\Phi(x) \in \mathbb{R}^{a \times m}$ has rank $a$, and that $a \times a$-matrix $M(g, x)$ satisfies (12) for all $k \in \mathbb{N} \cup \{0\}$, $x \in \mathcal{X}$ and $g \in \mathcal{G}$. Then,*

(i) $M(gh, x) = M(g, h \circ x)M(h, x)$ *for any $g, h \in \mathcal{G}$ and $x \in \mathcal{X}$.*
(ii) $M(g^\ell, x) = M(g, x)^\ell$ *for any $\ell \in \mathbb{Z}$, $g \in \mathcal{G}$, and $x \in \mathcal{X}$.*
(iii) $M(g, g^\ell \circ x) = M(g, x)$ *for any $\ell \in \mathbb{Z}$, $g \in \mathcal{G}$, and $x \in \mathcal{X}$.*

*Proof.* First note that, from (12) with $k = 0$, we have

$$M(g, x)\Phi(x) = \Phi(g \circ x) \tag{13}$$

for any $g \in \mathcal{G}$ and $x \in \mathcal{X}$.

Using (13) repeatedly, we have

$$M(gh, x)\Phi(x) = \Phi(gh \circ x) = \Phi(g \circ (h \circ x)) = M(g, h \circ x)\Phi(h \circ x) = M(g, h \circ x)M(h, x)\Phi(x).$$

The rank assumption of $\Phi$ proves (i).

Also, (13) implies $M(e, x) = id$ for the unit $e \in \mathcal{G}$. We will first prove (ii) and (iii) with $\ell > 0$.

For (ii), note that the repeated use of (12) necessiates

$$\Phi(g^\ell \circ x) = M(g, x)\Phi(g^{\ell-1} \circ x) = \cdots = M(g, x)^\ell \Phi(x).$$

On the other hand, $\Phi(g^\ell \circ x) = M(g^\ell, x)\Phi(x)$. Equating these two expression of $\Phi(g^\ell \circ x)$ proves (ii) with $\ell > 0$.

Meanwhile, from (12) we have $\Phi(g^{\ell+1} \circ x) = M(g, x)\Phi(g^\ell \circ x)$, while

$$\Phi(g^{\ell+1} \circ x) = \Phi(g \circ (g^\ell \circ x)) = M(g, g^\ell \circ x)\Phi(g^\ell \circ x).$$

This proves the assertion (iii) for $\ell > 0$.

Now, substituting $x \leftarrow g^{-1} \circ x$ for (iii) with $\ell = 1$, we obtain $M(g, x) = M(g, g^{-1} \circ x)$. On the other hand, substituting $h \leftarrow g^{-1}$ for (i), we get $M(g, g^{-1} \circ x)M(g^{-1}, x) = M(e, x) = id$. Thus,

$$M(g^{-1}, x) = M(g, x)^{-1}.$$

Replacing $g$ with $g^{-1}$ in (ii) thus leads to $M(g^{-\ell}, x) = M(g^{-1}, x)^\ell = M(g, x)^{-\ell}$ for any $\ell \in \mathbb{N}$. This shows that (ii) holds for the negative integers as well. Also, substituting $g \leftarrow g^{-1}$ in (iii) yields $M(g^{-1}, g^{-\ell} \circ x) = M(g^{-1}, x)$. Taking the inverse of the both sides proves the assertion (iii) for the negative integers. $\square$

*Proof of Proposition F.1.* Let $h, g \in \mathcal{G}$ be given. Since $\mathcal{G}$ is a connected commutative Lie group, the exponential map $\exp : \mathfrak{g} \to \mathcal{G}$ is surjective, where $\mathfrak{g}$ is the Lie algebra of $\mathcal{G}$ [19]. Therefore, there exists some $\eta \in \mathfrak{g}$ such that $\exp(\eta) = h$. Then, for any $n \in \mathbb{N}$, we can define $h^{\frac{1}{n}} := \exp(\eta/n)$ and $h^{\frac{1}{n}} \to e$ as $n \to \infty$.

By the uniform continuity assumption on $M(\cdot, x)$, for any $\epsilon > 0$, we can choose $n$ large enough so that

$$\|M(gh^{\frac{1}{n}}, g^{-n} \circ x) - M(g, g^{-n} \circ x)\|_F < \epsilon, \tag{14}$$

and

$$\|M(gh^{\frac{1}{n}}, h \circ x) - M(g, h \circ x)\|_F < \epsilon. \tag{15}$$

From Lemma F.2 (iii), we have

$$M(gh^{\frac{1}{n}}, g^{-n} \circ x) = M(gh^{\frac{1}{n}}, (gh^{\frac{1}{n}})^n g^{-n} \circ x),$$

and thus it follows from the commutativity assumption that

$$M(gh^{\frac{1}{n}}, g^{-n} \circ x) = M(gh^{\frac{1}{n}}, h \circ x). \tag{16}$$

At the same time, Lemma F.2 (iii) implies $M(g, g^{-n} \circ x) = M(g, x)$ so that (14) and (16) necessiates

$$\|M(gh^{\frac{1}{n}}, h \circ x) - M(g, x)\|_F < \epsilon. \tag{17}$$

Finally, the combination of (15) and (17) guarantees

$$\|M(g, h \circ x) - M(g, x)\|_F$$
$$\leq \|M(g, h \circ x) - M(gh^{\frac{1}{n}}, h \circ x)\|_F + \|M(gh^{\frac{1}{n}}, h \circ x) - M(g, x)\|_F < 2\epsilon.$$

Because $\epsilon > 0$ is arbitrarily small, $\|M(g, h \circ x) - M(g, x)\|_F = 0$ necessarily holds, and the claim follows. $\square$

**Proposition F.3.** *Suppose that, for a compact connected Lie group $\mathcal{G}$ and connected $\mathcal{X}$, $M : \mathcal{G} \times \mathcal{X} \to \mathbb{R}^{a \times a}$ in (12) satisfies the intra-orbital homogeneity, and that $\Phi(x) \in \mathbb{R}^{a \times m}$ has rank $a$ for all $x$. If $M(g, x)$ is continuous with respect to $x$, then $M(g, x)$ is similar to $M(g, x')$ for all $x, x'$; that is, there is some $P \in GL(a, \mathbb{R})$ such that $PM(g, x)P^{-1} = M(g, x')$ for all $g \in \mathcal{G}$.*

*Proof.* From Lemma F.2, we have

$$M(gh, x) = M(g, h \circ x)M(h, x).$$

Combining this with intra-homogeneity $M(g, h \circ x) = M(g, x)$ provides

$$M(gh, x) = M(g, x)M(h, x),$$

which means that, for each fixed $x$,

$$M_x : \mathcal{G} \to GL(a; \mathbb{R})$$

defined by $M(g, x) = M_x(g)$ is a representation of the Lie group $\mathcal{G}$ [19]. Now, if $\mathcal{G}$ is compact and connected as assumed in the statement, $M_x(g)$ is completely reducible, and $M_x$ is similar to a direct sum of irreducible representations. We then use the fact from character theory [19] that the multiplicity of any irreducible representation $D$ in $M_x$ can be computed by

$$\langle M_x | D \rangle = \int_{\mathcal{G}} tr(M(g, x)) \overline{tr(D(g))} \mu(dg), \tag{18}$$

where $\mu$ is a Haar measure of $\mathcal{G}$ with volume 1, and $\overline{tr(D(g))}$ is the complex conjugate of $tr(D(g))$. Because $\langle M_x | D \rangle$ is a multiplicity, it takes an integer value. At the same time, by its definition and the continuity of $M(\cdot, x)$, this value is continuous with respect to $x$. Thus, $\langle M_x | D \rangle$ must be constant on $\mathcal{X}$ by the connectedness of $\mathcal{X}$. That is,

$$\langle M_x | D \rangle = \langle M_{x'} | D \rangle$$

for all $x, x' \in \mathcal{X}$. This means that, irrespective of $x$, $M(g, x)$ is similar to the direct sum of the same set of irreducible representations, and the claim follows. $\square$