# OpenReview forum: "Unsupervised Learning of Equivariant Structure from Sequences"
_NeurIPS.cc/2022/Conference — NeurIPS 2022 Accept_

### Official Review · Reviewer_hTb6 · 2022-07-08

**Rating:** 5
**Confidence:** 2
**Soundness:** 3 good
**Presentation:** 3 good
**Contribution:** 2 fair

**Summary:**

The authors of this paper investigate the time series models with certain stationary properties, and use the learned symmetry structure for predicting the future in time series.  They consider the type of time series where each sequence is generated by initial observation and some fixed transition operator. Then they propose to optimize a homeomorphic function with equivariance property w.r.t. a group of transition operators. In the experiments, they evaluate the proposed approach on sequences generated based on 3 image datasets.The results show that their method can successfully learn transition operators such object rotation, hue rotation, and object translation. By applying these operators to the initial time step, they show that they could predict the future sequence for a few number of time steps.

**Questions:**

I have some questions:

- Can you motivate the ablation study in the experiments? I don’t understand why you are doing it.

- Regarding the comparison against SimCLR and CPC, I find it a bit hard to understand the experiments and interpret the results. Can you please provide more details?

- With noisy background in the image examples, do you find the optimization more difficult? Intuitively it adds challenges to learn the representations for the objects there, but I wonder how harder it becomes in your case, since you assume a purely static background.

General suggestion:

- I think it is interesting to study the equivariance property in time series in general. On the other hand, I feel it is also important to showcase the robustness / generalization when tackling trajectory prediction tasks. In other words, the point estimations are not that great in many use cases, and we would want to characterize the model uncertainty. So I wonder if you could do some reasoning about the prediction uncertainty in this proposed method.


**Limitations:**

To my best knowledge, the authors have comprehensively discussed the limitations of their work and the potential negative impact.

**Strengths And Weaknesses:**

Originality: In my opinion, this is an interesting paper because they try to investigate the equivariance property in stationary time series models. From this perspective this idea is novel, though I believe the problem defined in this paper can be alternatively tackled using existing methods on disentangled representation learning. (I will think of this weakness more as insignificance rather than originality, because in general studying equivariance in time series is a great topic, and it is not reasonable to argue about originality merely because some existing methods can solve the same type of problems.)

Significance: Regarding disentangled representation learning. I mainly wonder what we additionally achieved with this proposed method, compared to the existing methods on disentangled representation learning. The reason is that, the authors specifically consider time series with fixed transition operators, e.g. rotation, translation, and the goal is to predict the future observations in the sequences. Because of the strong assumption on fixed transition operators, these prediction tasks, in my opinion, can also be framed as generating examples by varying factors of variation using models in [1,2,3].
While the authors have explained how their work differs from the prior work on disentangled representation learning, I would appreciate it if they do some comparison against some of those methods such as [1,2,3].For both learning disentangled representations and generation tasks, those VAE-based methods will also achieve impressive results in datasets such as MNIST, FashionMNIST, Dsprites. On the other hand, this proposed method can only do point-estimation, while those VAE-based models can characterize the prediction uncertainty in a variational manner. This is my main concern about the significance of this work.

Quality: This paper is well structured in the sense that they provide proof of theory that supports their motivation and sufficient empirical evaluations as well.

Clarity: This paper is well written and clearly conveys the idea in the paper.

[1] Chen, Ricky TQ, et al. "Isolating sources of disentanglement in variational autoencoders." Advances in neural information processing systems 31 (2018).

[2] Kim, Hyunjik, and Andriy Mnih. "Disentangling by factorising." International Conference on Machine Learning. PMLR, 2018.

[3] Esmaeili, Babak, et al. "Structured disentangled representations." The 22nd International Conference on Artificial Intelligence and Statistics. PMLR, 2019.

---

> ### Author Response · Authors · 2022-08-02
> **Response to Reviewer htB6 (part1)**
>
> Thank you very much for the valuable comments. Below we respond to each concern / question raised in your review.
> Please see the last part (part3) of this response for the numbered references used in our comments.
>
> > I mainly wonder what we additionally achieved with this proposed method, compared to the existing methods on disentangled representation learning. in my opinion, can also be framed as generating examples by varying factors of variation using models in [1,2,3].
>
> Would you please let us know how VAE can be used to “predict” the future for any given sequence?  As far as we understand, the studies listed in the comment only study $p(x)$ and do not solve the future prediction task.
> One straightforward application of VAE to the sequential dataset in our paper would be to disregard the sequential relation in the dataset and train the generative model that can mimic the “unordered” version of the dataset while introducing some form of the regularization to identify the independent factor of variations. However, such a generative model does not seem to directly help predict the future of the sequence that requires the explicit form of the transition that includes speed/acceleration.
> The very recent topographic VAE(TVAE) [1] is one of the works with a similar goal that uses a VAE framework as well; however, unlike our approach, TVAE  engineers an explicit “cycle” framework in their model, and using VAE for the prediction task investigated in our study does not seem like a trivial task.
> We also note that we cannot cast the TVAE directly to compare on the sequential dataset we use in our dataset, because we deal with a set of sequences with varying speeds and different cyclic orders.
>
>
> Indeed, we also do consider it an important future work to develop a framework that can deal with future prediction with uncertainty, modeling $p({\bf s_p} | {\bf s_c})$ by leveraging the popular models like VAE, PixelCNN [2], Normalizing flows [3, 4], and recent diffusion probabilistic models [5].
> However, we believe that this is not within the scope of our current work, and as such, we do not believe it necessary to use VAE in our problem setting. Also, unlike VAE for which we need to carefully tune the hyperparameter to balance the KL term and the reconstruction term, our method does not require any hyperparameter that we have to cross-validate on each dataset.

---

> > ### Author Response · Authors · 2022-08-02
> > **Response to Reviewer htB6 (part2)**
> >
> > > Can you motivate the ablation study in the experiments? I don’t understand why you are doing it.
> >
> > Below we elaborate on the motivation behind each variant of our method used in our ablation study.
> > We added a similar description in Appendix as well.
> >
> >
> > *Fixed block models*
> >
> > In the pioneering work of [6] that endeavors to learn the symmetry in a linear system using the representation theory of commutative algebra, the authors hard-code the irreducible representations/block matrices in their model.  Our study is distinctive from many applications of representation theory and symmetry learning in that we uncover the symmetry underlying the dataset not by introducing any explicit structure but by simply seeking to improve the prediction performance. We therefore wanted to experiment on how the introduction of the hard-coded symmetry like the one in [6] would affect the prediction performance.
> > As we see in the result, introducing the hard-coded symmetry as an inductive bias does not positively affect the predictive performance, suggesting that, in truth, it can be unnecessary (if not harmful) to use a hard-coded symmetry in order to find the symmetry in the dataset.
> >
> > *Reconst models*
> >
> > We trained this model to show the importance of the use of the validation sequence ${\bf s_p}$.
> > In our default algorithm, we train our encoder and decoder with the prediction loss $\mathcal{L}^p$ in eq.(2) over the future horizon of length $T_p-T_c$ . We therefore wanted to verify what would happen to the learned representation when we train the model with the reconstruction loss  $\mathcal{L}^r$ in eq.(3), in which the model predicts the observations contained in the conditional sequence.
> > As we discuss in Section 6, the estimated transition with the Rec. model was inaccurate, possibly because the transition matrix overfitted to the conditional sequence-specific information, and the learned encoder was far from the equivariant map we aim to obtain. Similar results are reported in [7], in which the author showed that the latent structure of the predictive model well captured the cyclic structure of the sequences. We re-confirmed that the prediction of the future frames is important in learning the symmetry behind the sequence.
> >
> > *Neural*$M^*$
> >
> > Our method is *meta* in that we distinguish the internal training of $M^*$ for each sequence from the external training of the encoder $\Phi$. Put in another way, the internal optimization process of $M^*$ itself is the function of the encoder. To measure how important it is to train the encoder with such a *meta* approach, we evaluated the performance of Neural$M^*$ approach.
> > To reiterate, Neural$M^*$ uses a neural network $M^*_\theta$ that directly outputs $M^*$ on the conditional sequence, and trains the encoder and decoder via
> >
> > $ \sum_{t= T_c+1}^{T_c+T_p} ||  \Psi(M_\theta^* (\mathbf{s_c} )^{t-T_c} \Phi(s_{T_c})) - s_t ||_2^2 $
> >
> > thereby testing the training framework that is similar to our method *minus* the *meta* component.
> > Our experimental results show that the *meta* component is in fact very important in extracting the “global” symmetry that enables the inference of transition operators that acts on all element in the dataset in the same way (i.e., equivariance).
> >
> > *Neural transition*
> >
> > One important inductive bias that we introduce in our model is that we assume the latent transition to be linear. We therefore wanted to test what happens to the results of our experiment if we drop this inductive bias.
> > Namely, for $T_c=2$  we trained a 1DCNN that takes in the past two time frames to output the next frame.
> > As we can confirm in our experimental result, the linear inductive bias in the latent space has a positive effect on extrapolating the future.

---

> > > ### Author Response · Authors · 2022-08-02
> > > **Response to Reviewer htB6 (part3)**
> > >
> > > > Regarding the comparison against SimCLR and CPC, I find it a bit hard to understand the experiments and interpret the results. Can you please provide more details?
> > >
> > > We conducted this experiment to compare how well each method encodes the information of transition in each sequence. This is the reason why the regression targets are those that determine the transition (translation, color rotation, etc.).  Please recall that our model unwittingly recovers the hidden structure in the sequential dataset by simply training the model to be able to linearly predict the future.
> > > This experiment is a part of our investigation of how the uncovered hidden structures relate to the known parameters of the sequences, and how well the learned representation can be used to predict those parameters.
> > > We chose SimCLR and CPC as baseline methods simply because they are de-facto standards of deep unsupervised representation-learning methods.
> > > Following the convention in the standard evaluation of representation learning, we also added the result of regressing the digit classes for Sequential MNIST/MNIST-bg in the revision (Figure 9).  We can confirm in all results that our method performs competitively (and much better than SimCLR and CPC), and that the learned structures are useful in the downstream task as well.
> > > In the revised version of our paper, we updated the explanations for these comparative experiments.
> > >
> > > > With noisy background in the image examples, do you find the optimization more difficult? Intuitively it adds challenges to learn the representations for the objects there, but I wonder how harder it becomes in your case, since you assume a purely static background.
> > >
> > > We must say that it depends on the types of noise to be considered. If the background noise has some stationary property, our method possibly learns the dynamics. But if not, we may need some probabilistic models to model the noisy information (Because we have not tried both situations in our paper, we cannot tell how hard it is). Meanwhile, as we report in our paper,  the training with ImageNet background (MNIST-bg) did not seem to negatively affect the efficacy of our method.
> > >
> > > **References**
> > >
> > > [1] T. Keller and M. Welling. Topographic vaes learn equivariant capsules. NeurIPS, 2021.
> > >
> > > [2] A. Van den Oord, et al. Conditional image generation with pixelcnn decoders. NeurIPS, 2016.
> > >
> > > [3] D. Rezende and S. Mohamed. Variational inference with normalizing flows. ICML, 2015.
> > >
> > > [4] I. Kobyzev, S. JD Prince, and M. A. Brubaker. Normalizing flows: An introduction and review of current methods. TPAMI, 2020.
> > >
> > > [5] J. Ho, A. Jain, and P. Abbeel. Denoising diffusion probabilistic models. NeurIPS, 2020.
> > >
> > > [6] T. Cohen and M. Welling. Learning the irreducible representations of commutative lie groups. ICML, 2014.
> > >
> > > [7] T. Keller and M. Welling. T. Keller and M. Welling. Predictive coding with topographic variational autoencoders. ICCV workshop, 2021

---

> ### Author Response · Authors · 2022-08-09
> **We would appreciate your feedback!**
>
> Dear Reviewer htB6,
>
> We have made a detailed reply to your comments and revised our manuscript to reflect your concerns.
> In our rebuttal comments, we discussed the relationship between VAE and our work. Also, we described the detailed setting on the comparative experiments against SimCLR and CPC.
> We also noted the motivation behind each variant of our method and added the description to the revised manuscript in the appendix D.1.
> We would appreciate it if you give us further feedback on the revision and our rebuttal.

---

### Official Review · Reviewer_XtDt · 2022-07-10

**Rating:** 5
**Confidence:** 2
**Soundness:** 3 good
**Presentation:** 2 fair
**Contribution:** 2 fair

**Summary:**

This paper proposes to discover structural properties of the data by solving a sequential prediction tasks.
The idea is that solving the prediction tasks produces equivariant models, which by guarantees from representation theory, means that the learned representations are simultaneously block-diagonalized, with blocks corresponding to disentangled factors.

Specifically, the paper learns an encoder $\Phi$, a decoder $\Psi$, and a matrix-valued function $M$, such that given adjacent observations $s_t, s_{t+1}$, the difference $\||\Psi (M\Phi(s_t)) - s_{t+1}\||_2^2$ is minimized. The training borrows idea from meta learning, where the algorithm first estimates $M$ based on a part of the sequence (denoted $\mathbf{s}_c$), and $\Phi$ is then estimated using this $M$ and the rest of the sequence (denoted $\mathbf{s}_p$).

Empirical results show that 1) the proposed method is able to train on only adjacent time frames and extrapolate to unseen steps in the future, and that 2) the learned linear transitions can be simultaneously block-diagonalized, with blocks corresponding to 1 factor of variation.

**Questions:**

Questions:
- The choice of $M$ seems to rely on knowledge of the true dynamics. How to obtain such knowledge in general? How robust is the method to "model mismatch", e.g. when internal optimization is performed on sequences with noises or with varying velocity? How can the model be extended to handle some amount of model mismatch?
- Would the choice of $T_c, T_p$ affect the performance?
    - Choosing $T_c=2$ should be sufficient if the data perfectly follows the model assumption (e.g. constant velocity), but this may not be the case when there's model mismatch.
- Would the proposed method be able to handle multiple objects?
- Sec 5.2, swapping $M^*$: my understanding is that $M^*$ should depend on the sequence-specific action $g$. However in Fig 4(a), the two sequences clearly have different actions (e.g. the bottom row seems to be doing both color rotation and shape rotation, while the top row only has shape rotation) and different orbits (since the digits are not the same). How could $M^*$ be the same then?
- Fig 8 in the appendix: line 255 in the main text says that the proposed method gives "better representations" but it's unclear how so. Please explain the success/failure modes of the proposed method and the reasons.
- Line 276, about simultaneously block-diagonalizable: please provide reference for this.


Clarifications:
- Line 124: is $\circ_{latent}$ different from $\circ$?
- Line 220: "digit4 only training": does it mean that the encoder/decoder have not seen other digits during training?
- Line 233: what's the reason for splitting $H$ into 8 (and not some other number of) subtensors?
- Line 250: what does "fixed block structure" mean? Does dividing $H$ into 8 subtensors count at fixing the block structure?
- Line 303: what does "sufficiently large" mean for $m$?


Writing: overall I find the paper a bit hard to parse, possibly due some inconsistent notations and my own in-familiarity to the topic.
Some concrete things:
- Please fix citation [13].
- Line 115: do you mean members of $S$ (rather than $s$, which is itself a member of $S$)?
- Line 143: maybe consider a different notion than $M(g, s_1)$, since $M$ has been defined to be a single-argument function from $\mathcal{G}$ to $\mathbb{R}^{a \times a}$.
- Line 179: should it be "optimized on $s_p$"?
- Inconsistent notations:
    - $H_{+1} H_{+0}^\dagger$ in eq (4) and $H_2H_1^\dagger$ in Fig 1.
    - $T_p$ on line 227, $t_p$ on line 245.
- Line 233 & 234: it would be more consistent to use either 256 or $m$ (but not both) in defining $H, H_k$.
- The paragraph on line 252: it would be better to briefly explain what "better representation" means.
- Line 289: there seems to be a typo in the definition of $\hat{M}^*$.


**Limitations:**

The paper discusses the limitations of the proposed method, such as failing to work with non-invertible transitions, e.g. ShapeNet dataset.

The paper also discusses potential societal impact if the method were to employed in real-world sequential prediction problems.

**Strengths And Weaknesses:**

Strengths
- It's an interesting idea to learn disentangled representations by the simple prediction task.
- The paper presents a variety of empirical results and is clear on implementation details.
- The paper provides hypotheses on some empirical results (Sec 6) and is open about the limitations of the proposed methods, e.g. failing to work on non-invertible transitions.

Concerns
- [Clarified in the response] While inductive biases are necessary as established by prior work, the proposed method seems too tailored to the specific data structure, and it seems unfair to compare with baseline models that do not have such knowledge about the data.
- [Draft updated] The messages from some empirical results could be made clearer; please see detailed questions below.
- [Clarified in the response] Please discuss the relation to the line of work about incorporating physics into the prediction model, e.g. PhyDNet (Guen & Thome 20), and disentangled representation learning in videos in general.

=== **Update** ===

The authors' responses have address most of my concerns. Many of my previous questions were because I didn't get the main point the paper, since I had trouble parsing the paper as I mentioned in comments on clarity.
The writing is now improved and my review has been updated accordingly.

---

> ### Author Response · Authors · 2022-08-02
> **Response  to Reviewer XtDt (Part1)**
>
> Thank you very much for spending time to review our paper.  Below we respond to each concern and comment in your review.
> Please see the last part (part4) of this response for the numbered references used in our comments below.
>
> > it seems unfair to compare with baseline models that do not have such knowledge about the data.
>
> On the contrary, all the baseline models we compare against our method in the experiment section  (Neural M*, Rec Model, Neural Transition) also assume that the transitions are stationary in each sequence and seek a time-invariant operator to predict the future, so that the comparisons are therefore made on the same background of prior knowledge.  In particular, all but Neural Transition assume that the latent of each sequence transitions with a sequence-specific linear operator $M$ in the way of
> $\Phi(s_{t+1}) = M \Phi(s_{t})$, just as assumed in our proposed model.  Neural Transition assumes the model $M(\Phi(s_{t-1}), \Phi(s_{t})) = \Phi(s_{t+1})$ with a possibly nonlinear function $M$.  This transition model too is based on the same prior knowledge.
>
> For the motivations and more detailed discussion on the ablation studies, please see our response to htB6. Also, because we are computing $M^*$ by Least Square regression, by construction our method is robust against some degree of homoscedastic fluctuation in the speed as well.
> We would also like to emphasize that one message we would like to convey in this paper is that we can take advantage of the continuity of the time series dataset to automatically capture the hidden structure of the dataset by simply aiming to improve the future prediction performance.
>
> The situation in which we can obtain a set of short stationary sequences is also not irregular as well, because any continuous sequence with sufficient time-resolution is always piece-wise stationary in the way of local Taylor approximation.
>
> > Please explain why the connection to meta-learning is necessary or how it can help provide more insights.
>
> In our paper, we are experimenting with what we call “Neural $M^*$” in order to answer this question. In the “Neural $M^*$” version of our method, we are updating the $M^*$ and $\Phi$ simultaneously without constructing the explicit inner loop.
> This is in contrast to our main proposed approach that solves the inner optimization completely and directly backpropagates the change in $M^*$ induced by the change in $\Phi$, which is optimized in the outer loop.
> The comparison between “Neural $M^*$” and our main proposed method suggests that there is something important about this *meta* aspect in acquiring the extrapolation capability.
> We believe that further investigation of this phenomenon is the most important agenda in our future works.
>
>
> > Please discuss the relation to the line of work about incorporating physics into the prediction model, e.g. PhyDNet (Guen & Thome 20), and disentangled representation learning in videos in general.
>
> The “disentanglement” discussed in PhyDNet is significantly different from the “disentanglement” in our paper, both in terms of purpose and nature.
> PhyDNet disentangles the “dynamics that can be explained with a certain form of PDE” from the residual dynamics.  PhyDNet also engineers its architecture for the purpose of achieving this “disentanglement”.
> Hamilton generative network (HGN) [5] is another instance of the physics-inspired prediction model. It tries to encode sequences into (abstract) momentum and position variables.
> As another form of disentanglement, [6] as well as [7] take the approach of separating the time-invariant component from time-variant component.
>
> Meanwhile,  the disentanglement that is discussed in our paper is more relevant to those discussed in [8,9],  which pertains to the algebraic decomposition of the transition operators in the field of representation theory [8,10] and symmetry learning [11,12].
>
> Also, we would like to emphasize that, unlike many studies with a primary focus on disentanglement,  we do not have any specific design of disentanglement in our framework;  as we report in our work,  the disentanglement emerges as the byproduct of our endeavor in training the model to predict the (possibly very short) time sequence through meta-learned linear latent operators.
> As we discuss in the theory section, this surprising coincidence is in alignment with the theory of group representation and equivariance.
> This is in strong contrast to previous representation learning methods for video and the family of studies whose primary goal is the disentanglement of some specific form, as we are discovering the disentangled structure without explicitly seeking it.

---

> > ### Author Response · Authors · 2022-08-02
> > **Response to Reviewer XtDt (Part2)**
> >
> > > The choice of $M$ seems to rely on the knowledge of true dynamics.  How robust is the method to model mismatch?
> >
> > We do not assume anything about the true dynamics other than the stationary properties, such as constant acceleration / constant velocity over short frames (as short as three). Such an assumption is natural when the dynamics is continuous. [1,2] also takes the similar strategy of splitting the time sequences into shorter time frames over which the structure of the dataset is stationary (The latter for example uses such sub-sequences of length as long as 16).  We indeed expect our model to fail in prediction if the future velocity/acceleration changes dramatically.
> > We would also want to emphasize that the goal of our study is not to propose a general method of video prediction, but to point out the novel connection between extrapolation ability and the sequential/algebraic data structure, which were previously discussed based on hard-coded inductive bias.  As the most important consequence, we are able to infer from each sequence a  transition operator that causes a consistent effect on all other sequences (i.e. equivariance).
> > Please See [3], for example, for previous efforts to identify the structure in sequential dataset.
> >
> >
> > > Would the choice of $T_c,T_p$ affect the performance?
> > >> Choosing $T_c=2$ should be sufficient if the data perfectly follows the model assumption (e.g. constant velocity), but this may not be the case when there's model mismatch.
> >
> > Indeed, we observed that using longer prediction horizon $T_p$ or longer $T_c$ generally improves the performance.  Also, on noisy sequential datasets  (e.g. when  the velocity of angles are sampled from gaussian distribution), using larger $T_c$ should be particularly effective since the internal least square algorithm absorbs the noises when $T_c$ is large. However, as we showcase in our manuscript, $T_c=2$ and $T_p=1$ sufficed for competitive performance. This implies that our extrapolation objective is uncovering the representation-theoretic symmetry in the dataset from mere triplets.
> >
> > > Would the proposed method be able to handle multiple objects?
> >
> > We have been conducting preliminary experiments and we are seeing some promising results with additional modules; however, because the prediction itself is not the main focus of this study, we excluded our preliminary results from the scope of our manuscript.
> >
> >
> > > Sec 5.2, swapping $M^*$: my understanding is that  $M^*$ should depend on the sequence-specific action. However in Fig 4(a), the two sequences clearly have different actions (e.g. the bottom row seems to be doing both color rotation and shape rotation, while the top row only has shape rotation) and different orbits (since the digits are not the same). How could  be the $M^*$  same then?
> >
> > First, let us clarify that the actions applied in the pair of sequences in Figure 4a are exactly the same; the velocities of hue angle and rotation angle are shared across the pair. Because of the definition of the HSV space, the change in the value of the hue might not be in agreement with the human perception: For example, one might see the gradient around yellow w.r.t the hue value is steeper than the gradient around the red (like in our example in Figure 4a).  Also, it was difficult to clearly perceive the changes in color in the original version of the figure, because we were shading the images in the $T_c$ part of the sequence to distinguish them from the $T_p$ part of the sequence.  We therefore removed the shade for better visibility. We are sorry for the lack of clarity on this matter.
> > We would like to note that the most important fruit of our work is the way to infer $M^*$ that is exclusively dependent on $g$; indeed, our inferred $M^*$ seems to be dependent on $g$ but independent of the choice of $s_1$.
> > As an important consequence, we can apply one $M^*$ inferred from one sequence to another sequence in a consistent manner.
> > We investigate in our theory (Section 3.1) the mechanism underlying this acquired equivariance property.
> >
> >
> > > Fig 8 in the appendix: line 255 in the main text says that the proposed method gives "better representations" but it's unclear how so. Please explain the success/failure modes of the proposed method and the reasons.
> >
> > Because we were talking about the ability of our model to predict the transition parameters from the estimated $M^*$, we used the word “wellness” here in the context of representation learning. We removed the ambiguous expression in the revision. We also added the result of linearly regressing the digit classes in Appendix, and our representation performs competitively on this task as well (Figure 9).
> > However, as we mention in our response to all reviewers, the primary merit of our study is not about downstream tasks. We presented the results in Figure 8 to study how our representation relates to the structural factors of the dataset. Please also see our response to Reviewer dgWX.

---

> > > ### Author Response · Authors · 2022-08-02
> > > **Response to Reviewer XtDt (Part3)**
> > >
> > > > Line 276, about simultaneously block-diagonalizable: please provide reference for this.
> > >
> > > We added [4] as a reference to the simultaneously block-diagonalization. Please see Section 1.2 Representation Theory for the details.
> > >
> > >
> > > > Line 124: is $\circ_{latent}$ different from $\circ$?
> > >
> > > We would like to point out that $\circ$(action in the observation space) is different from $\circ_{latent}$(action in the latent space), and we are sorry if this notation was confusing. We denote the action on input space by the binary operation $\circ: G \times X -> X$ while we denote the action on the latent space by $\circ_{latent}: G \times R^{a\times m} \rightarrow R^{a\times m}$. In the revised manuscript, we define the latent space action without using \circ_{\rm latent} to avoid the confusion.
> > >
> > >
> > > > Line 143: Maybe consider a different notation than $M(g, s)$
> > >
> > > We are sorry that this notation was confusing.  We were originally hoping to distinguish the transition operator in the ideal model from the “estimated transition operator” corresponding to the group element $g$ by using the subscripted $M_g$ to refer to the former and using $M(g,s)$ to refer to the “transition operator estimated from the sequence that begins with s and transitions with $g$“.
> > > We revised Section 3 to include the description of the difference between $M(g,s)$ and $M_g$.
> > >
> > > > Line 220: "digit4 only training": does it mean that the encoder/decoder have not seen other digits during training?
> > >
> > > Yes. The models only observed the examples of digit 4 during the training of the encoder/decoder. For the evaluation of downstream regression task, we trained a linear classifier on feature space for each model using the training sets containing all digits, and evaluated the regression error ($R^2$ score) on the test sets containing all digits.
> > >
> > >
> > > > Line 233: what's the reason for splitting $H$ into 8 (and not some other number of) subtensors? Line 250: what does "fixed block structure" mean? Does dividing $H$ into 8 subtensors count at fixing the block structure?
> > >
> > > As we report in our manuscript, the set of $M^*$s obtained from the latent variable that were trained to well-predict the future in our framework can be simultaneously block-diagonalized, and each block has a theoretical connection with the irreducible representation.  We emphasize, however, that this simultaneous block-diagonalizability was not achieved by some explicit algorithm or mechanism to learn it, but by simply training the model to be able to well-predict the future. We therefore wanted to experiment what happens if we introduce the inductive bias of the block-diagonal structure and train the network so that $M^*$ from each sequence will be a direct sum of a fixed number of 2x2 blocks (Say $M_i*$; i=1,..8).  When $M^*$ is assumed to take such a form, learning $M^*$ acting on $\mathbb{R}^{2 \times 256}$ dimensional tensor is  equivalent to learning $M_i^*$ acting on $2 \times 256$ dimensional tensor for each $i=1,...,8$.
> > > In the pioneering work of [8] that learns the symmetry in a linear system, the authors hard-code these blocks (See equation (2) of [8]). Our result suggests that, when the objective function is appropriately constructed, such a structured inductive bias might be unnecessary.  We also discuss the motivation of our ablation study in our response to the Reviewer htB6.
> > >
> > > > Line 303: what does "sufficiently large" mean for $m$?
> > >
> > > We are sorry for the lack of explanation. If $m$ is less than $a$,  we cannot obtain the pseudo inverse because of the rank deficient in $\Phi(s_{t-1}) \Phi(s_{t-1})^{\rm T}$. Thus $m$ should be at least larger than $a$. We added this explanation to the revised paper.

---

> > > > ### Author Response · Authors · 2022-08-02
> > > > **Response to Reviewer XtDt (Part4)**
> > > >
> > > > *About Typographies*
> > > >
> > > > Thank you very much for pointing out our typos. We have fixed the errors as below:
> > > >
> > > > > Please fix citation [13]”
> > > >
> > > > Thank you, we fixed the citation.
> > > >
> > > > > Line 115: do you mean members of $S$ (rather than $s$ which is itself a member of $S$)?
> > > >
> > > > Yes, thanks for pointing this out. We meant to say “members of $S$”.
> > > >
> > > > > Line 179: should it be "optimized on ${\bf s_p}$?:
> > > >
> > > > Thank you very much.  This is indeed a typo. We fixed it.
> > > >
> > > > > Inconsistent notations: ...
> > > >
> > > > To reduce confusion with $H_{+1}$, we stopped to use $H_i$ to denote $\Phi(s_i)$.  We also fixed the typo of $t_p$.
> > > >
> > > > > Line 233 & 234: it would be more consistent to use either 256 or m (but not both)...
> > > >
> > > > We fixed the notation accordingly.
> > > >
> > > > > The paragraph on line 252: it would be better to briefly explain what "better representation" means.
> > > >
> > > > We made sure to reiterate that we intended here to measure the “wellness” of the representation by predicting multiple structural features in the dataset (See Figure 8).
> > > > We also added in the revision the result of regressing the digit class (Figure 9).
> > > >
> > > > > Line 289:  there seems to be a typo in the definition of $\hat{M}^*$
> > > >
> > > > Thank you very much, this is a typo and we fixed it.  We should have had $A(\hat{M}^*):=$ in place of $\hat{M}^*:=$ here.
> > > >
> > > > **References**
> > > >
> > > > [1] A. Hyvarinen, and H. Morioka. Unsupervised feature extraction by time-contrastive learning and nonlinear ica. NeurIPS, 2016.
> > > >
> > > > [2] S. Zhang, Y. Wang, and A. Li. Cross-view gait recognition with deep universal linear embeddings. CVPR, 2021.
> > > >
> > > > [3] T. Keller and M. Welling. Topographic vaes learn equivariant capsules. NeurIPS, 2021.
> > > >
> > > > [4] R. Kondor. Group theoretical methods in machine learning. Columbia University, 2008.
> > > >
> > > > [5] P. Toth, et al. Hamiltonian generative networks. ICLR, 2020.
> > > >
> > > > [6] J. Hsieh, et al. Learning to decompose and disentangle representations for video prediction. NeurIPS, 2018.
> > > >
> > > > [7] R. Kabra, et al. Simone: View-invariant, temporally-abstracted object representations via unsupervised video decomposition. NeurIPS, 2021.
> > > >
> > > > [8] T. Cohen and M. Welling. Learning the irreducible representations of commutative lie groups. ICML, 2014.
> > > >
> > > > [9] L. Falorsi, et al. Explorations in homeomorphic variational auto-encoding. arXiv preprint arXiv:1807.04689, 2018.
> > > >
> > > > [10] S. H. Weintraub. Representation Theory of Finite Groups: Algebra and Arithmetic, volume 59. American Mathematical Society, 2003.
> > > >
> > > > [11] A. Zhou, T. Knowles, and C. Finn. "Meta-learning symmetries by reparameterization." ICLR, 2021.
> > > >
> > > > [12] N. Dehmamy, et al. "Automatic Symmetry Discovery with Lie Algebra Convolutional Network." NeurIPS, 2021.

---

> > > > > ### Comment · Reviewer_XtDt · 2022-08-08
> > > > > **Thank you for your detailed responses**
> > > > >
> > > > > Thank you very much for the detailed responses and clarifications. I've adjusted my review, but since I missed the key point of the paper previously, I want to make sure that my understanding is now correct:
> > > > > - The reason that the prediction task gives disentanglement is that by representation theory, an equivariant model gives features that can be simultaneously diagonalized.
> > > > > - Even though the model cannot yet be proven equivariant theoretically (i.e. Prop 3.1 & 3.2 are related to orbits), empirically the model appears to be equivairant and discovers disentangled factors.
> > > > >
> > > > > If the above understandingly is correct, then perhaps one way to make the main message clearer is to highlight this connection a bit more and de-emphasize the meta learning aspect (since it's a simple training trick without theoretical justification).

---

> > > > > > ### Author Response · Authors · 2022-08-09
> > > > > > **Thank you very much for understanding our points**
> > > > > >
> > > > > > Thank you very much for your comment, and we are glad our response clarifies your concerns.
> > > > > >
> > > > > > > The reason that the prediction task gives disentanglement is that by representation theory, an equivariant model gives features that can be simultaneously diagonalized.
> > > > > >
> > > > > > Yes, if the model is equivariant, the representation theory guarantees that $M^*$ can be simultaneously block-diagonalized across different orbits by an appropriate change of basis matrix $U$. Using such $U$, the original feature is transformed to $U\Phi(s_t)$, for which each block of $UM^*U^{-1}$ acts on the corresponding subspace of $U\Phi(s_t)$ exclusively.
> > > > > >
> > > > > >
> > > > > > > Even though the model is not shown to be equivariant theoretically (i.e. Prop 3.1 & 3.2 are related to orbits), empirically the model is found to be equivariant.
> > > > > >
> > > > > > Yes, although Prop 3.1 and 3.2 together show that $M^*(g, x)$ with the same $g$ are similar (in the sense of linear algebra) on different orbits, they do not fully prove the equivariance because the "equivariance relation" requires that $M^*(g, x)$ with the same $g$ do not depend on $x$ at all.
> > > > > > And yes, the full equivariance is achieved by our proposed meta-learning algorithm.
> > > > > >
> > > > > >
> > > > > > > highlight this connection a bit more and de-emphasize the meta learning aspect (since it’s a simple training trick without theoretical justification).
> > > > > >
> > > > > > In our current manuscript, we have some emphasis on the meta-learning aspect because, as we empirically demonstrate in ablations (Section 5), this aspect seems to be important in achieving the equivariance that cannot be justified with Thm 3.1 and Thm 3.2 alone.
> > > > > >
> > > > > > We shall note that the fact that the equivariance relation has a direct connection to disentanglement is a well-known fact [Reference 4, 8 in our initial response]; the achieved disentanglement supports our claim that our model is successfully learning an equivariant model.
> > > > > > At the same time, identifying the hidden equivariance relation in the dataset is still a big challenge today,
> > > > > > and our theory shows that we can make the model “almost equivariant” by training the model to be able to predict the future with linear transition in the latent space.
> > > > > > However, the empirical evidence shows that, with our “meta” framework, the equivariant model can be learned, and we show in our ablation study that the  “meta” part of our framework is important (In section 5, Neural $M^*$ corresponds to the non-meta version of our training procedure).
> > > > > > To clarify this point, we modified the end of our introduction by reordering the statements and changing some of the expressions.

---

> > > > > > > ### Comment · Reviewer_XtDt · 2022-08-09
> > > > > > > **Thank you for the further clarifications**
> > > > > > >
> > > > > > > Thank you for the further clarifications and the updates to the draft.

---

### Official Review · Reviewer_dgwX · 2022-07-15

**Rating:** 6
**Confidence:** 3
**Soundness:** 3 good
**Presentation:** 2 fair
**Contribution:** 3 good

**Summary:**

The paper studies the problem of learning symmetry from sequential data with a certain stationarity property. In particular, from time sequences of length at least three, meta-learning is used to learn representations such that a future observation can be predicted well by a linear transition. It is shown that doing so can tease out the disentangled structure by simultaneous block-diagonalization. Each block then corresponds to a disentangled feature.

The meta-learning framework is very straightforward: We learn an injective mapping \phi for each time step, such that a linear transition relates time step t to time step t+1. The loss function involves a decoder that maps the linear transition \phi_t-1 to \phi_t. The transition can be parameterized in various complex ways, but as mentioned above only a linear map is used. The inner loop consists of minimizing the prediction loss, while the outer loop updates the encoder. The main assumption is that the velocity/acceleration is preserved within each observation. Experiments on Sequential MNIST, 3D Shapes, and SmallNORB show the efficacy of the proposed method.




**Questions:**

See above:

Minor comments:

There are quite a few typos and somewhat awkward sentence constructions. Below are a few examples:
- Typo, line 16: *in machine learning?
- Line 20: should be "has succeeded"
- Line 34-35: "in the way of meta-learning" -> by meta-learning/by the way of meta-learning/by means of meta-learning
- Line 44-45: "There are rich literatures in unsupervised/weakly supervised" -> There is a rich literature...
- Line 63: "Many studies impose algebraic constraints that reflects some form of geometrical assumption." -> that reflect some form of..
- Line 86: identifiebility -> identifiability



**Limitations:**

Yes, adequately addressed.

**Strengths And Weaknesses:**

Strengths:
- The basic idea of the paper is quite simple: That we can learn some underlying algebraic structure in an unsupervised way when the transitions can be represented linearly in some latent space. It follows the line of work on Slow Feature Analaysis, ICA and more recent work by Keller and Welling, and also seems reminiscent of work on Koopman analysis (although multiple dynamics are considered here). The paper proposes to use meta-learning to predict a future observation and model only linear transitions. The meta-learning setup is also quite simple.
- The experiments are carried out on reasonable datasets: Sequential MNIST, 3D Shapes and SmallNorb. The overall results and quality of ablation is good.
- The extension to constant acceleration is also reasonable, as are the discussion of limitations.

Weaknesses:
- The paper is actually quite hard to read. I think it can be made really good with a smoother presentation. Even though the main idea is quite simple, I often had difficulty following the details and trying to decode what the authors might have meant.
- The experiments are good in the sense that they cover some good datasets, and have good ablations. However, I have difficulty in placing their significance, given the authors only report measures such as equivariance error. Can we have another downstream task in which such learnt representations are shown to be helpful? If not, is it possible to have some reasonable baselines to get a sense of how good the variants proposed here are? Overall I like the main idea -- it is simple and elegant, but I am not quite sure what to make of the results, and I think the paper could do a better job at presenting them.

---

> ### Author Response · Authors · 2022-08-02
> **Response to Reviewer dgwX**
>
> Thank you very much for your valuable comments. We corrected the typographies and inconsistency of notations in the revision.
>
> As for the concern regarding the downstream task, we show in the figure 8 in the appendix section the results of linearly regressing the true transition parameters of the sequences from $M^*$. Because we are computing $M^*$ with $T_c =2$ in most of our experiments, the $M^*$s used in this evaluation are regressed from a pair of latent variable; therefore, all the predictors used for Fig 8 are functions of $[\Phi(s_1), \Phi(s_2)]$, and we are providing the result of regressing the transition parameters from $[\Phi(s_1), \Phi(s_2)]$  (For SimCLR and CPC evaluation in this experiment, we are directly regressing the parameters from $[\Phi(s_1), \Phi(s_2)]$ ).
> Also, in the revision we also added our result of linearly regressing the digit class of $s$ from $\Phi(s)$, and our representation performs better than the representation learned by SimCLR and contrastive predictive coding (CPC). Please see Figure 9 for more details about the results.
>
> At the same time, it is generally not straightforward to determine how the representation should be evaluated, and there is no unique method to justify the evaluation methods.  It is because we wanted to study the  *wellness* of representation from multiple viewpoints that we added the regression results of Fig 8 in addition to the equivariance error, which we believe measures the direct merit of our study by quantifying how well we learned the hidden global symmetry from the local training of each individual sequence.

---

> > ### Comment · Reviewer_dgwX · 2022-08-08
> > **Reponse**
> >
> > Thanks for the comments. I have gone through the updated PDF and see a number of changes from the intro to the related works (including more references on Koopman, which I think are relevant), description, experiments. (As an aside, I think there is a typo in figure 6). I again looked at the appendix based on the suggestions by the authors, and do feel the presentation has seen some improvement. Although I still feel there might still be quite some room for improvement. I will raise my score by a notch.

---

### Author Response · Authors · 2022-08-02
**Response to all reviewers**

Dear all reviewers,

Thank you very much for spending time reviewing our work. As suggested by the reviewers, we fixed the typographies and the notational problems in the revision to improve the clarity of our writing.
We respond to the questions and concerns raised by each reviewer in an individual response to each reviewer.

One main message that we would like to convey in this paper is that we have found a meta-type unsupervised framework that can uncover the symmetry of the dataset when trained to extrapolate the future well on different sequential datasets.
The uncovered symmetry takes the form of disentangled features and their property is in strong alignment with the group representation theory. Our method is distinctive from many past studies in that we do not engineer any specific framework to extract the symmetry/disentangled representation; the symmetry emerges naturally from the objective designed solely for the purpose of training a model that can predict the future linearly in the latent space.
In this regard, we believe that the value of our study is not very much about the sheer ability of our proposed method to predict the future or to disentangle the features, but about the novel connection between meta-type training, extrapolation, and algebraic symmetry including those pertaining to disentanglement.

Also, in Figure 6, we updated the results of a 1st-order version of our proposed method; the result in the original manuscript was the outcome of a wrong snapshot model (premature model). We therefore replaced the results with the outcome of the models at the end of training. The replacement, however, does not change the claim of our paper; the second-order model version achieved the best extrapolation performance in this experiment.

---

### Meta-Review · Area_Chair_qE2h · 2022-08-24

**Recommendation:** Accept
**Confidence:** Less certain

**Metareview:**


While there was a certain lack of enthusiasm in the scores of the reviewers, the author's answers cleared the concerns of the reviewers participating in the discussion and overall the recommendation leans towards acceptance. This paper is, in the reviewers' opinions, sound and adds to the literature on unsupervised learning of symmetry. The formulation (of learning symmetry by only modelling linear transitions) is nicely simple. Experiments and evaluations generally were considered of adequate quality.



**Award:**

No

---

### Decision · Program_Chairs · 2022-09-14

Accept